# APlaud: Adaptive Personalized Low-Rank Decomposition for User-Specific LLM

## Abstract

In this paper, we introduce and study the problem of *personalized survey response prediction* using fine-tuned large language models (LLMs). This task poses unique challenges: limited per-user training data, scalability of model storage, and the need to exploit shared survey structures. To address these issues, we propose **APlaud** (Adaptive Personalized Low-rank and User-specific Nested Decomposition), a lightweight and scalable framework for LLM personalization. APlaud extends the LoRA paradigm by separating adaptation into a frozen, shared low-rank basis and a compact user-specific correction, augmented with a rank-one residual for finer personalization. To further reduce per-user parameter cost and mitigate overfitting, the correction matrix can be factorized into an even lower-rank form. Empirical results demonstrate that APlaud achieves efficient, scalable personalization across users while outperforming state-of-the-art LoRA-based personalized LLM approaches in both generalization and inference efficiency.

## 1 Introduction

Surveys and polls such as the Pew Research Survey (61), the General Social Survey (GSS) (70), the Gallup World Poll (20), and the American National Election Studies (ANES) (2) have long been indispensable for informing public policy, advancing social science, and guiding marketing decisions (26; 78; 54; 10; 18). Beyond these canonical examples, fields such as marketing research, product design, social and political science, biomedicine and psychology, and education all rely heavily on surveys and interviews as primary tools for understanding human perspectives. Yet these instruments now face mounting challenges, including rising costs, declining response rates, and persistent concerns over accuracy and representativeness (43; 11; 44). Driven by surging demand in the multi-billion-dollar global survey and market-research sector, researchers and practitioners are increasingly turning to synthetic participants – LLM-generated respondents – as a scalable alternative for augmenting or partially replacing traditional data collection (5; 40; 1; 33; 16; 28; 65; 53; 63; 36; 45; 7; 72; 91; 71; 29; 4). Industry adoption has accelerated rapidly: the Qualtrics 2025 marketing trend report (64) highlights synthetic responses as a direct substitute for human respondents, while enterprises such as YouGov and Kantar, along with startups including SyntheticUsers (73), OpinioAI (58), Delve.ai (15), and PersonaLive.ai (60), have begun offering synthetic survey responses at scale, underscoring the growing momentum of synthetic subject modeling across both research and commercial applications.

However, the majority of existing research (primarily from the domain sciences) has focused on directly applying LLMs with simple prompting strategies that condition responses on basic demographic attributes—a setting we refer to as **persona-level (subpopulation) prediction** (5; 1; 33; 67; 47; 4; 68; 83; 42; 22; 24; 79). Such studies typically test whether LLMs inherently capture the correct distribution of answers for a target demographic subpopulation. Results consistently show that LLM-generated responses often exhibit cultural and in-group/out-group biases, lack demographic nuance, and tend to produce homogenized opinions with reduced diversity and greater predictability compared to real human participants (68; 83; 47; 42; 22; 24; 79; 56).

More recently, a small but growing body of work has sought to mitigate these limitations by aligning persona-level response distributions (e.g., demographics, socioeconomic status, or ideology) with empirical human data through fine-tuning or reinforcement learning with human feedback (RLHF) (38; 90; 71). Yet these approaches remain constrained to the subpopulation level, leaving

open the critical challenge of modeling survey response personalization at the level of individual users.

**The Personalized Survey Response Prediction Problem.** To move beyond subpopulation-level modeling and address personalization at the level of individual users, we formally introduce and investigate the *personalized survey response prediction problem*: *Can a fine-tuned (personalized) LLM replicate, predict, or simulate an individual's responses to new survey questions, given their answers to a previous set of questions in an existing survey?* This problem has immediate real-world relevance. In both public and private sectors, organizations frequently conduct large-scale surveys and retain ownership of the resulting data. When new, related questions arise, it is natural to want to follow up with the original participants. Yet re-contacting respondents is often costly, time-consuming, or infeasible. As a result, stakeholders are increasingly turning to LLMs to generate synthetic responses as a preliminary step before committing resources to new data collection (64).

This problem lies at the heart of the emerging notion of *digital twins*, where the goal is to simulate user behavior or preferences in a way that reflects individual-specific characteristics rather than merely subpopulation-level distributions (73; 60). It also connects directly to ongoing advances in recommendation and personalization, where the development of *personalized LLMs*—or *personalized alignment*—has become a central objective (90; 27). Our work is further motivated by direct industry collaborations that revealed a critical operational bottleneck: organizations frequently need to pose follow-up questions to respondents who are unavailable or prohibitively costly to re-contact due to attrition, survey fatigue, or escalating incentive expenses. Personalized synthetic respondents offer a direct solution to this problem by enabling practitioners to ask new questions to modeled representations of original respondents, thereby preserving continuity of analysis without repeated recruitment. The overarching aim of this line of research is to align model outputs with the preferences, stylistic tendencies, and behavioral patterns of specific individuals. Yet, to the best of our knowledge, personalized LLMs have not been systematically developed or evaluated for individual-level survey response prediction—a critical and largely unexplored opportunity that our work seeks to address.

Finally, from a benchmarking perspective, individual-level survey response prediction provides a rigorous and interpretable testbed for evaluating both digital twins and personalized LLMs. While benchmarks such as LaMP (66) assess personalization across tasks such as citation generation, tagging, product rating, and title creation, survey response prediction offers a complementary, domain-specific benchmark focused on capturing users' latent preferences, behavioral patterns, and opinions. To our knowledge, this work is the first to explicitly formulate and study the survey prediction problem in the personalized LLM setting.

## 1.1 RESEARCH CHALLENGES AND OUR APPROACH

The main challenges of the problem are threefold. First, the number of survey questions in existing datasets and real-world scenarios is typically modest—ranging from tens to, at most, a few thousand. This means the amount of personalized data available per user is often much smaller than the size of the personalized parameters, which can easily lead to severe overfitting. Second, the number of users can be extremely large—ranging from thousands to tens of millions in industry-scale applications. Even though PEFT methods like LoRA reduce the number of trainable parameters relative to full fine-tuning, each per-user adapter can still involve hundreds of millions of parameters (albeit a small fraction of the base model). At scale, maintaining such adapters quickly becomes prohibitively expensive in terms of both storage and deployment, making per-user fine-tuning impractical. Third, surveys typically ask the same set of questions across users, which naturally induces shared semantics and correlations. An effective personalization strategy should exploit this inherent structure, rather than treating each user entirely in isolation.

To address these challenges, we propose **APlaud**, a framework for developing truly personalized LLMs by leveraging existing survey questions and responses. Our approach is designed to balance scalability, data efficiency, and structural exploitation. An overview of the proposed framework for the personalized survey response prediction problem is shown in Figure 1.

The key idea of APlaud is to utilize the shared space introduced by the standard LoRA training $AB$, and then apply singular value decomposition (SVD) to uncover the orthogonal decomposition $U\Sigma V$. We hypothesize that the subspaces $U$ and $V$ are relatively stable across users, enabling us to introduce a small personalized matrix $C_u$ to capture individual user preferences. Specifically, APlaud builds on

the LoRA paradigm by decoupling adaptation into a frozen, shared low-rank basis and a compact, user-specific residual correction. Concretely, we decompose a LoRA update via SVD into orthogonal matrices $U$, $\Sigma$, and $V$, and inject a learnable user-specific matrix $C_u$, yielding an adapted weight of the form

$$W = W_0 + U(\Sigma + C_u)V^\top,$$

where $U$ and $V$ are shared across users while $C_u$ encodes personalization. To further enhance expressiveness, APlaud augments this formulation with a small, user-specific low-rank residual term. Finally, to reduce parameter overhead and mitigate overfitting, the personalized matrix $C_u$ is itself factorized into a lower-rank form $P_u Q_u$. This design directly addresses the three key challenges identified earlier: (1) compact user-specific parameters mitigate overfitting when per-user data is limited; (2) low-rank factorization further minimizes per-user storage and parameter overhead; and (3) shared subspaces maximize the utilization of semantic information common across users.

**Our Contributions.** The contributions of this work are threefold:

- We introduce the *personalized survey response prediction problem*, a novel and practically important task that bridges survey research and the development of personalized LLMs, and propose it as a new benchmark for personalization.
- We present **APlaud**, a scalable and parameter-efficient framework that combines shared LoRA-derived subspaces with lightweight, user-specific corrections via SVD-based decomposition and residual terms.
- We provide extensive empirical evaluation showing that APlaud reduces the per-user parameter cost of state-of-the-art personalized LLM methods (e.g., OPPU) by orders of magnitude, while achieving comparable or superior predictive accuracy.

In real operational settings, our method provides survey companies, market-research organizations, and enterprise stakeholders with a scalable and cost-efficient alternative to traditional respondent workflows. Concretely, our framework alleviates key deployment pressures by enabling organizations to: (i) generate reliable follow-up responses without re-contacting participants; (ii) substantially reduce recruitment, incentive, and panel-maintenance costs; (iii) rapidly prototype, iterate on, and validate new survey instruments; and (iv) extend panel longevity through persistent, personalized synthetic respondents calibrated to each user's historical responses.

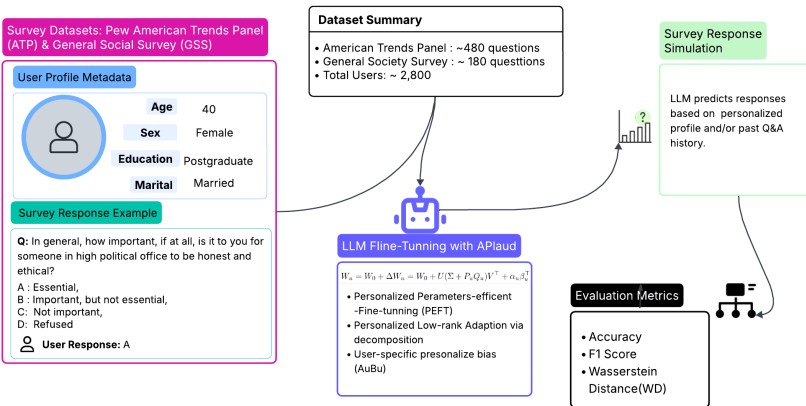

Figure 1: Overview of **APlaud** for personalized survey response prediction. The framework combines shared LoRA-derived subspaces with user-specific low-rank corrections to achieve scalable and data-efficient personalization.

## 2 PRELIMINARY: LORA AND PERSONALIZED LLM

### 2.1 LORA AND ITS VARIANTS

LoRA (Low-Rank Adaptation) is one of the most widely adopted parameter-efficient fine-tuning (PEFT) techniques (35). It is motivated by the low intrinsic dimensionality hypothesis (3), which

suggests that fine-tuning can often be effectively performed in a lower-dimensional subspace. LoRA achieves this by introducing low-rank updates to the dense layers of a pre-trained neural network, instead of modifying the full parameter matrix.

Formally, let $W_0 \in \mathbb{R}^{d \times k}$ denote the original weight matrix of a dense layer. LoRA introduces a trainable update $\Delta W \in \mathbb{R}^{d \times k}$ such that the updated layer is parameterized by:

$$W = W_0 + \Delta W.$$

Rather than learning $\Delta W$ directly, LoRA factorizes it as the product of two low-rank matrices:

$$W = W_0 + sAB,$$

where $A \in \mathbb{R}^{d \times r}$, $B \in \mathbb{R}^{r \times k}$, and $r \ll \min\{d, k\}$. The matrix $A$ is typically initialized with a Gaussian distribution, while $B$ is initialized to zero. The scalar $s$ is a scaling factor used to stabilize and control the magnitude of the update.

This factorization significantly reduces the number of trainable parameters from $d \times k$ (in full fine-tuning) to $r \times (d + k)$ in LoRA, offering a substantial efficiency gain. For simplicity of exposition, we assume in the remainder of this paper that $W_0$ is a square matrix (i.e., $d = k$), which is commonly the case in transformer-based architectures. Nonetheless, our method generalizes naturally to non-square matrices.

We note that several recent approaches have explored the use of Singular Value Decomposition (SVD) to enhance LoRA training. For example, given the SVD of a pre-trained weight matrix $W_0 = U\Sigma V^\top$, one can initialize the LoRA adapters using the factorized components—setting $A = U\Sigma^{1/2}$ and $B = \Sigma^{1/2}V^\top$—to provide a data-informed starting point for fine-tuning.

Some methods adopt SVD-inspired parameterizations directly during training. For instance, AdaLoRA (89) approximates the weight update matrix as $\Delta W = P\Sigma Q^\top$, where $P$ and $Q$ are constrained to be approximately orthogonal via regularization terms $\|P^\top P - I\|$ and $\|Q^\top Q - I\|$, and $\Sigma$ is a learnable diagonal matrix. PiSSA (Principal Singular Values and Singular Vectors Adaptation) (55) initializes LoRA adapters using the top singular components of the pre-trained weights via SVD while freezing the remaining components. This strategy improves convergence speed and accuracy by aligning updates with the most informative subspace. In contrast, MiLoRA (81) and KASA (80) propose to freeze the top singular components and instead fine-tune the minor singular directions, emphasizing complementary subspaces for adaptation. To the best of our knowledge, our approach (Aplaud) is the first to leverage SVD-based decomposition strategies to support personalized LLM adaptation and personalized alignment.

## 2.2 PERSONALIZED LLM

**Prompt-based Personalization.** User information—such as demographics, preferences, behavioral signals, and historical activity – derived from user-generated content or contextual background, is typically encoded into prompts (87; 1; 5; 51; 6; 19; 48; 88; 50). When user history is extensive, techniques such as prompt refinement (49) and retrieval-augmented generation (RAG) (82) can be employed to construct more informative and scalable prompts.

In the context of personalized survey response prediction, the number of available questions and responses per user typically ranges from a few dozen to a few thousand–sufficient to fit within the context window of commercial LLMs such as ChatGPT. However, when using open-source models with more limited context capacity, it may be necessary to summarize prior interactions or apply RAG-based mechanisms to generate compact, user-specific prompts.

**Encoding-based Personalization.** In this class of approaches, user data and preferences are compressed into vector representations or embeddings (57; 49; 69), which are then integrated into the model to modulate token-level processing and output generation for personalization. Similarly, user-specific latent variables and reward models have been developed to enable personalization through reinforcement learning (62; 25; 9).

While these methods allow for individualized conditioning, they typically rely on a shared transformer backbone, resulting in a uniform inference architecture across all users. This shared structure implicitly assumes a common "thinking process" for all individuals, which may be too restrictive to accurately capture the full range of human variability in preferences, reasoning patterns, and response styles.

**Parameter-based Personalization.** In this category, the first class of methods encodes user preferences directly into model parameters via full-parameter personalization, where a separate model is trained for each user by fine-tuning (41; 49; 85) or optimizing via reinforcement learning (38; 86) all model weights. While offering maximal flexibility, this approach is often prohibitively expensive in both storage and computation. The second class of methods leverages parameter-efficient fine-tuning (PEFT), which introduces per-user adaptation modules—such as LoRA, while keeping the base model frozen (74; 14; 37). Next, we will introduce OPPU (One PEFT Per User) (75), which is the current SOTA per-user LLM framework and provides a straightforward way to the personalized survey response prediction problem.

**OPPU for Personalized Survey Response Prediction.** OPPU (One PEFT Per User) (75) builds an independent parameter-efficient fine-tuning (PEFT) model for each user. In practice, this often entails assigning each individual their own LoRA (Low-Rank Adaptation) module (35). Formally, the parameterization can be expressed as

$$W_u = W_0 + s_1 AB + s_2 A_u B_u,$$

where $W_0$ denotes the pretrained model weights, $AB$ is a shared low-rank adaptation trained on the entire dataset using standard LoRA (first stage, with $s_1$ to its scaling factor), and $A_u B_u$ represents the user-specific low-rank parameters. In this setup, the shared component $AB$ captures global adaptation across all users, while personalization is introduced in a second stage by training the individual-specific parameters $(A_u, B_u)$ on top of the updated weights $W_0 + AB$ (with $s_2$ as its scaling factor).

Note that a follow-up study (74) extends OPPU by allowing target users to select and assemble personalized PEFT modules from a shared pool using their historical data. This reduces storage costs by avoiding one fully independent PEFT per user, but it sacrifices accuracy relative to fully personalized models. Other approaches have proposed reinforcement learning to incorporate user-specific preferences via reward models (90; 27). However, even with these advances, a foundational challenge remains: *how to represent each user within the LLM architecture in a way that is parameter-efficient, storage-optimized, structure-aware, and resistant to overfitting given the limited data available for each individual?*

## 3 APlaud Approach

**APlaud** (*Adaptive Personalized Low-rank and User-specific Nested Decomposition*) is a novel parameter-efficient fine-tuning method designed to personalize large language models (LLMs) at the per-user level. It extends the standard LoRA framework by enabling scalable and expressive user-specific adaptation while minimizing the per-user parameter footprint. Figure 2 illustrates the APlaud method in comparison with LoRA, which provides only global (non-personalized) adaptation.

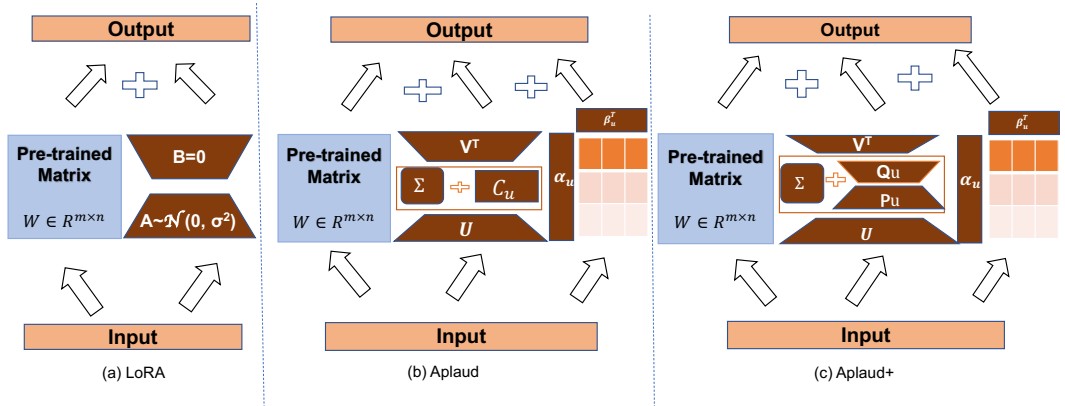

Figure 2: LoRA vs. APlaud Overview

### 3.1 Model Details: Compact Corrections and Residual Refinement

To address the limitations of existing personalized LLMs such as OPPU (74), which assigns each user an independent pair of matrices $(A_u, B_u)$, APlaud instead reuses the shared low-rank update

$AB$ learned in the first stage of a standard LoRA adaptation. Specifically, we apply singular value decomposition (SVD) to the global LoRA update:

$$\Delta W = AB = U\Sigma V^\top,$$

where $U \in \mathbb{R}^{d \times r}$ and $V \in \mathbb{R}^{d \times r}$ are orthogonal matrices capturing dominant update directions, and $\Sigma \in \mathbb{R}^{r \times r}$ is a diagonal matrix of singular values $(\sigma_1, \ldots, \sigma_r)$.

Intuitively, each singular vector $V_i$ in $V$ corresponds to a semantic direction against which the input $x$ is measured. The projection $V_i^\top x$ is scaled by the singular value $\sigma_i$ and then mapped to the corresponding output direction $U_i$, yielding a transformed coordinate $\sigma_i(V_i^\top x)U_i$. Because user-specific datasets are small and the semantic structure of survey questions is relatively stable across users, we hypothesize that the shared subspaces $U$ and $V$ capture most of the semantic directions needed for personalization.

Building on this, instead of training new $(A_u, B_u)$ for each user, we inject a compact, user-specific correction matrix $C_u \in \mathbb{R}^{r \times r}$ into the singular spectrum:

$$W_u = W_0 + \Delta W_u = W_0 + s\,U(\Sigma + C_u)V^\top,$$

where $W_0$ is the base model weight and $s$ is a scaling factor. The shared $U$ and $V$ are fixed across all users, while $C_u$ provides personalized adjustments. Importantly, $C_u$ is extremely lightweight: for rank $r = 8$, it requires only 64 parameters, and even for rank $r = 64$, only 4096, on the order of a single singular vector.

Compared with $U\Sigma V^\top$, the modified form $U(\Sigma + C_u)V^\top$ remains in the subspace spanned by $U$, i.e.,

$$U(\Sigma + C_u)V^\top x \in \mathrm{Span}(\mathrm{Col}(U)).$$

However, $C_u$ enables each user to reweight and mix semantic directions in $V$, thereby reflecting their individual preferences and importance weights.

To capture fine-grained, idiosyncratic nuances beyond the shared subspace, APlaud augments the representation with a lightweight personalized residual:

$$W_u = W_0 + s\,U(\Sigma + C_u)V^\top + \alpha_u\beta_u^\top,$$

where $\alpha_u, \beta_u \in \mathbb{R}^d$ are learned per-user vectors. This rank-one residual enables APlaud to adjust beyond the shared low-rank structure, modeling distinctive patterns that cannot be expressed solely within the subspace spanned by $U$ and $V$. In principle, the residual can be extended to higher rank, but we find that a rank-one correction is generally sufficient in our experimental settings (see Appendix for detailed results).

**Aplaud+: Nested Low-Rank Factorization.** To further compress the user-specific component and improve regularization, we factorize the correction matrix as

$$C_u \approx P_u Q_u,$$

yielding the personalized update

$$W_u = W_0 + sU(\Sigma + P_u Q_u)V^\top + \alpha_u\beta_u^\top,$$

where $P_u, Q_u \in \mathbb{R}^{r \times k}$.

APlaud and Aplaud+ drastically reduce the per-user parameter footprint. For APlaud, with rank $r = 64$, the correction matrix $C_u$ requires only $64 \times 64 = 4096$ parameters. For Aplaud+, using a nested inner rank of $k = 16$, the factorization $P_u Q_u$ requires only $2rk = 2048$ parameters per user for each weight matrix $W$. The residual vectors $\alpha_u, \beta_u \in \mathbb{R}^d$ introduce an additional $2d$ parameters. By contrast, OPPU requires $2dr = 524{,}288$ parameters when $d = 4096$. Thus, APlaud and Aplaud+ achieve approximately $128\times$ and $256\times$ parameter reduction, respectively, without residual terms, and still over $42\times$ and $50\times$ reduction, respectively, when including the residual terms – all while preserving expressive capacity.

## 3.2 TRAINING PROCEDURE FOR APLAUD

The training procedure for **APlaud** consists of two stages, similar to other personalized LLM frameworks such as OPPU (75).

**Stage 1: Global Adaptation with LoRA-style Training.**
We begin with a standard parameter-efficient fine-tuning (PEFT) procedure such as LoRA, applied across the full training dataset comprising all users' responses. Specifically, we learn a global low-rank update:

$$W = W_0 + \Delta W = W_0 + AB,$$

where $W_0 \in \mathbb{R}^{d \times d}$ is the pre-trained weight matrix, and $A \in \mathbb{R}^{d \times r}$, $B \in \mathbb{R}^{r \times d}$ are trainable low-rank matrices. Following LoRA convention, $A$ is initialized from a standard Gaussian distribution and $B$ is initialized to zero, ensuring the pretrained model behavior is preserved at initialization. This stage captures population-level adaptation trends.

After training, we compute the singular value decomposition (SVD):

$$AB = U\Sigma V^\top,$$

where $U, V \in \mathbb{R}^{d \times r}$ and $\Sigma \in \mathbb{R}^{r \times r}$. These components define a shared low-rank subspace, which remains fixed for all users in the personalization stage.

**Stage 2: Personalized Fine-tuning of $C_u$ and Residual Terms.**
For each user $u$, we fine-tune a compact correction matrix $C_u$ together with residual vectors $\alpha_u, \beta_u$. We initialize $C_u = \mathbf{0}$, set $\alpha_u \sim \mathcal{N}(0, I)$, and $\beta_u = \mathbf{0}$. To stabilize training, we normalize $\alpha_u$ and introduce a scaling factor $m$, yielding:

$$\boxed{W_u = W_0 + s\,U(\Sigma + C_u)V^\top + m\frac{\alpha_u}{\|\alpha_u\|}\,\beta_u^\top.}$$

Here, $s$ controls the global scaling, while $m$ modulates the strength of the residual correction.

## 3.3 APLAUD+ TRAINING PROCESS

In **Aplaud+**, the correction matrix $C_u$ is further factorized into a nested low-rank form $P_u Q_u$. Training proceeds in three substages:

**Stage 2(a): Training $C_u$.** We first learn the full correction matrix $C_u \in \mathbb{R}^{r \times r}$ using user $u$'s data:

$$W_u = W_0 + s\,U(\Sigma + \gamma C_u)V^\top,$$

where $\gamma$ is a scaling factor similar to LoRA.

**Stage 2(b): Low-Rank Factorization of $C_u$.** Next, we compress $C_u$ via SVD:

$$C_u = U_C \Sigma_C V_C^\top.$$

Truncating to a smaller rank $k \ll r$, we initialize:

$$P_u = U_C \Sigma_C^{1/2}, \quad Q_u = \Sigma_C^{1/2} V_C^\top,$$

and re-train using user $u$'s data:

$$W_u = W_0 + s\,U(\Sigma + P_u Q_u)V^\top.$$

**Stage 2(c): Residual Learning.** Finally, we fine-tune residual vectors $\alpha_u, \beta_u$:

$$\boxed{W_u = W_0 + s\,U(\Sigma + P_u Q_u)V^\top + m\frac{\alpha_u}{\|\alpha_u\|}\,\beta_u^\top.}$$

**Remarks.** These substages serve distinct purposes: Stage 2(a) help initialize $P_u, Q_u$ from a learned $C_u$; Stages 2(b) and 2(c) then train the nested low-rank and residual components. While they could in principle be trained jointly, we find that separating them improves stability. Despite the per-user

independence of these stages, the small size of survey datasets (20–40 responses) allows training each user's model in just 1–2 minutes on a single A100 GPU.

**Personalized Parameter Size Summary.** - For APlaud, each user is represented by a correction $C_u \in \mathbb{R}^{r \times r}$ plus an optional rank-one residual $\alpha_u \beta_u^\top$, for a total of $r^2 + 2d$ parameters per weight matrix. - For Aplaud+, we use $P_u \in \mathbb{R}^{r \times k}, Q_u \in \mathbb{R}^{k \times r}$ with residual vectors, for a total of $2rk + 2d$ parameters per weight matrix.

Thus, APlaud and Aplaud+ provide fine-grained personalization with dramatically reduced per-user memory footprint, leveraging shared $U, V, \Sigma$ for structure while adapting lightweight corrections and residuals for individual flexibility.

# 4 EXPERIMENT

**Datasets** For survey data, we utilize data from two prominent sources of US public opinion: the annual Pew American Trends Panel (ATP) and the General Society Survey (GSS) (8)(70) for LLM simulation of survey and opinions(68)(71). We further evaluate our methods on LAMP Movie-Tagging(66), a publicly available personalized dataset from a non-survey domain to guarantee dataset diversity. The ATP is an annual longitudinal survey conducted by the Pew Research Center, based on a nationally representative panel of approximately 10,000 US participants recruited over multiple years, many of whom respond to repeated survey waves. From this dataset, we selected four specific waves that cover a diverse range of public opinion topics, including gender and leadership, trust in science, family and relationships, and economic inequality. For the GSS, we focus on Panel 20, a longitudinal cohort that provides rich repeated measurement data on topics such as political trust, social norms, religiosity, and inequality. This panel enables the study of temporal dynamics in US social attitudes and is particularly well-suited for evaluating user-specific LLM adaptation over time.

To explore user-level personalization and simulate LLM "ownership", we focus on the most engaged participants. Specifically, we retain respondents with at least 10 valid answers and remove "Refused" responses. Since each wave contains over 100 ASK-ALL items, the filtered dataset provides roughly 50 answered questions per person on average—sufficient for robust personalization and learning stable user embeddings (More details in Appendix A.2). After this filtering, we selected 200 users with the highest response rates and validated their responses across approximately 130 survey questions. To align the model output with the preferences and behavioral tendencies of the individual user, we first identify a subset comprising 30% of questions that capture key aspects of user personality and behavioral traits. This subset is used to construct a user-specific profile through LLM-based prompting. Then we split the rest of the question into three sections, 80% for training, 10% for validation, 10% for testing. This setup enables evaluation of each model's ability to simulate personalized responses with minimal supervision. For LAMP Movie Tagging dataset, we follow the work of OPPU to choose the top 100 users and also split the data with 8:1:1 ratio.

**Baseline** We compare our proposed method, APlaud, with several set of baseline approaches, including: (1) Non-personalized LoRA and its variatns: LoRA(34), PiSSA(55), MiLoRA(81), AdaLoRA(89) and QLoRA(17); (2) GPT-5 with profile only as best zero shot baseline; (3) Retrival-based approach, where we also use GPT-5 and augments the prompt with the five most relevant historical QA pairs, inserted as few-shot exemplars; (4) Personalized per user LLM: such as OPPU(75). We employ both Mistral-7B-v0.2-Instruct (39) and LLaMA-2-7B (76) as backbone models to verify the robustness of our approach under different base architectures.

**Evaluation Metrics** We evaluate model performance using three complementary metrics. Accuracy measures the proportion of correctly predicted survey responses compared to ground truth, providing a direct indicator of prediction reliability. F1 Score (macro) captures the balance between precision and recall, particularly useful when evaluating multi-class or imbalanced response distributions. Wasserstein Distance (WD)(see corresponding results in appendix) quantifies the distributional difference between the predicted and actual answer distributions, offering a fine-grained assessment of how closely the model captures user-specific response patterns.

**Experimental Settings** Due to space limitations, we leave more experimental details to the appendix. Our Code is available at `https://anonymous.4open.science/r/ICLR2026_Aplaud-4EEF/README.md`

**Experimental Performance** We first report our main results in Table 1 and Table 2. We also report the Relative of Improvement Results using one of our methods(Aplaud+) in Table 13 and Table 14 in

Table 1: Performance comparison across different datasets with llama2-7B backbone. Bold numbers indicate the best results within each dataset. Dataset names: G&L = ATP Gender & Leadership, TS = ATP Trust in Science, F&R = ATP Family and Relationships, EI = ATP Economic Inequality, GSS = General Social Survey. LAMP MV = LAMP Movie Tagging

| Method | G&L | | TS | | F&R | | EI | | GSS | | LAMP MV | |
|---|---|---|---|---|---|---|---|---|---|---|---|---|
| | ACC | Macro-F1 | ACC | Macro-F1 | ACC | Macro-F1 | ACC | Macro-F1 | ACC | Macro-F1 | ACC | Macro-F1 |
| **Non-Personalized** | | | | | | | | | | | | |
| LoRA | 0.6393 | 0.6140 | 0.7052 | 0.5343 | 0.5772 | 0.3348 | 0.4617 | 0.3308 | 0.3785 | 0.2479 | 0.6214 | 0.5076 |
| PiSSA | 0.6296 | 0.6142 | 0.7386 | 0.5620 | 0.5473 | 0.3427 | 0.4783 | 0.3722 | 0.3559 | 0.2333 | 0.6201 | 0.5280 |
| MiLoRA | 0.6714 | 0.6634 | 0.7406 | 0.5768 | 0.5551 | 0.3503 | 0.4618 | 0.3522 | 0.3836 | 0.2681 | 0.6308 | 0.5341 |
| AdaLoRA | 0.6700 | **0.6675** | 0.7779 | 0.5965 | 0.5564 | 0.3602 | 0.4716 | 0.3689 | 0.3907 | 0.2834 | 0.6146 | 0.5058 |
| QLoRA | 0.6618 | 0.6475 | 0.7467 | 0.5316 | 0.5408 | 0.3411 | 0.4683 | 0.3296 | 0.3738 | 0.2531 | 0.6302 | 0.5253 |
| **Personalized** | | | | | | | | | | | | |
| GPT5-profile | 0.5394 | 0.5340 | 0.6170 | 0.3448 | 0.4954 | 0.2879 | 0.4566 | 0.3491 | 0.4689 | 0.2570 | 0.5478 | 0.4507 |
| GPT5-RAG | 0.6377 | 0.6306 | 0.7001 | 0.6169 | 0.6174 | 0.3542 | **0.5213** | 0.4117 | **0.5106** | **0.3520** | - | - |
| OPPU | 0.6651 | 0.6559 | 0.7548 | 0.6275 | 0.6096 | 0.3612 | 0.5008 | 0.4091 | 0.3701 | 0.2588 | 0.6336 | 0.5147 |
| Cu | 0.6651 | 0.6453 | 0.7497 | 0.6423 | 0.5994 | 0.3475 | 0.4975 | 0.3611 | 0.3870 | 0.2590 | 0.6414 | 0.5358 |
| Aplaud | 0.6731 | 0.6581 | 0.7761 | 0.6637 | 0.6151 | 0.3669 | 0.5042 | 0.3945 | 0.3912 | 0.2715 | 0.6442 | 0.5366 |
| Aplaud+ | **0.6828** | 0.6642 | **0.7974** | **0.6824** | **0.6381** | **0.3704** | 0.5183 | **0.4180** | 0.3969 | 0.2719 | **0.6593** | **0.5529** |

Table 2: Performance comparison across different datasets with Mistral-7B backbone. Bold numbers indicate the best results within each dataset. Dataset names: G&L = ATP Gender & Leadership, TS = ATP Trust in Science, F&R = ATP Family and Relationships, EI = ATP Economic Inequality, GSS = General Social Survey. LAMP MV = LAMP Movie Tagging

| Method | G&L | | TS | | F&R | | EI | | GSS | | LAMP MV | |
|---|---|---|---|---|---|---|---|---|---|---|---|---|
| | ACC | Macro-F1 | ACC | Macro-F1 | ACC | Macro-F1 | ACC | Macro-F1 | ACC | Macro-F1 | ACC | Macro-F1 |
| **Non-Personalized** | | | | | | | | | | | | |
| Lora | 0.6554 | 0.6342 | 0.7072 | 0.4186 | 0.5681 | 0.2372 | 0.4708 | 0.3356 | 0.4336 | 0.2624 | 0.6669 | 0.5035 |
| PiSSA | 0.6521 | 0.6466 | 0.7375 | 0.5169 | 0.5408 | 0.2387 | 0.4525 | 0.3119 | 0.4110 | 0.2832 | 0.6801 | 0.5114 |
| MiLoRA | 0.6586 | 0.6499 | 0.7446 | 0.5039 | 0.5539 | 0.2572 | 0.4633 | 0.3297 | 0.4532 | 0.3146 | 0.6823 | 0.5014 |
| AdaLoRA | 0.6425 | 0.6318 | 0.7071 | 0.4768 | 0.6005 | 0.3653 | 0.4708 | 0.3114 | **0.5649** | **0.3573** | 0.6635 | 0.5349 |
| QLoRA | 0.6505 | 0.6368 | 0.7183 | 0.5127 | 0.5422 | 0.2486 | 0.4700 | 0.3307 | 0.4435 | 0.2903 | 0.7047 | **0.5517** |
| **Personalized** | | | | | | | | | | | | |
| GPT5-profile | 0.5394 | 0.5340 | 0.6170 | 0.3448 | 0.4954 | 0.2879 | 0.4566 | 0.3491 | 0.4689 | 0.2570 | 0.5478 | 0.4507 |
| GPT5-RAG | 0.6377 | 0.6306 | 0.7001 | 0.6169 | 0.6174 | 0.3542 | **0.5213** | **0.4117** | 0.5106 | 0.3520 | - | - |
| OPPU | 0.6731 | 0.6560 | 0.7852 | 0.6962 | 0.6368 | 0.3654 | 0.4925 | 0.3936 | 0.4322 | 0.2906 | 0.6917 | 0.4401 |
| Cu | 0.6828 | 0.6691 | 0.7781 | 0.6338 | 0.5746 | 0.2796 | 0.5008 | 0.3868 | 0.4548 | 0.2917 | 0.6982 | 0.5014 |
| Aplaud | 0.6828 | 0.6707 | 0.7852 | 0.7127 | 0.6278 | 0.3842 | 0.5042 | 0.3921 | 0.4506 | 0.3385 | **0.7159** | 0.5116 |
| Aplaud+ | **0.6876** | **0.6742** | **0.7862** | **0.7204** | **0.6537** | **0.4369** | 0.4992 | 0.3859 | 0.4605 | 0.3459 | 0.7081 | 0.5204 |

Appendix C.9. Aplaud families demonstrates markedly superior personalization capability across different datasets. Typically, on the Llama2-7B backbone, Aplaud+ delivers consistent gains of 6–10% in ACC and 11–15% in Macro-F1 over state-of-the-art non-personalized PEFT methods, and outperforms even SOTA general-purpose model (GPT5-profile) by 17.2% ACC and 33.2% Macro-F1 on average. Compared with the strong personalized adapter-based baseline OPPU, Aplaud also outperforms with a significant margin, improving 4.6% in ACC and 4.5% in Macro-F1. On the Mistral-7B backbone, the improvements is also pronounced, reaching 8–10% ACC and 24–36% Macro-F1 over non-personalized methods; surpassing retrieval-based approach (GPT5-RAG) 2.4% ACC and 7.8% Macro-F1 and outperform OPPU by 2.5% ACC and 10.2% Macro-F1.

Firstly, personalized approaches on average consistently surpass their non-personalized counterparts in terms of both accuracy (ACC) and macro-F1, highlighting the benefits of modeling user-specific adaptation. While adaptive non-personalized methods such as AdaLoRA and MiLoRA demonstrate relatively competitive performance, they remain inferior to personalized strategies on most datasets.

When comparing with stronger zero-shot general-purpose models and retrieval-based approaches (GPT-5 profile / GPT-5 RAG), our methods (Aplaud and Aplaud+) also demonstrate clear superiority. For example, in F&R data and with Mistral-7B as backbone, Aplaud+ achieves as large as 51.8% on Macro-F1, compared with zero-shot GPT and also 23.3% compared with GPT-RAG. This disparity reflects a fundamental limitation of retrieval-based personalization: Although free of training, the quality of each generated response relies on a narrow subsample of history records, which inevitably cannot capture a respondent's full behavioral signature, thus lacking personalization expressiveness. Especially in survey prediction tasks, a few retrieved samples rarely reflect the full spectrum of a user's attitudes and easily omit key signals. In contrast, our lightweight adapter architecture accumulates user-specific signals across the full interaction history and encodes them as persistent parametric memory, forming a holistic and stable representation of user preferences. This enables long-term

Table 3: Parameter count comparison per user per layer. We assume using Mistral-7B as foundation model and all LoRA-based PEFT with rank 64. For the SVD step, we retain the top 16 dimensions and bias term with rank 1.

| | Per-user per layer private parameter Calculation | #params | Percentage |
|---|---|---|---|
| OPPU | 4096×64×2×2 + (4096×64 + 1024×64)×2 +(4096×64 + 14336×64)×3 | 5,242,880 | 100% |
| SVD | (64×16×2)×7 | 14,336 | 0.27% |
| Aplaud | (64×64)×7 + (4096×1×2×2 + (4096×1 + 1024×1)×2 + (4096×1 + 14336×1)×3 | 110,599 | 2.11% |
| Aplaud+ | (64×16×2)×7 + 4096×1×2×2 + (4096×1 + 14336×1)×3 | 96,263 | 1.84% |

personalization rather than episodic conditioning, explaining why Aplaud/Aplaud+ substantially outperform retrieval-based baselines despite using far smaller models.

Compared with the representative personalized benchmark OPPU, our proposed Aplaud framework achieves consistent improvements across most datasets, yielding more balanced gains in both ACC and macro-F1. Notably, the enhanced variant Aplaud+ establishes new state-of-the-art performance on the majority of datasets, with average ACC improvements 4.6%, 2.5% and Macro-F1 improvements 4.5%, and 10.2% on Llama2-7B and Mistrial-7B, respectively. These results underscore that Aplaud not only advances beyond non-personalized tuning but also surpasses existing personalized approaches such as OPPU, thereby demonstrating the effectiveness of incorporating user-specific signals to enhance robustness and generalization across diverse domains.

Taken together, these results demonstrate the effectiveness of our proposed method: while existing non-personalized PEFT approaches capture generalizable knowledge, incorporating user-level personalization (as in Aplaud and Aplaud+) leads to more robust and balanced performance across diverse datasets.

**Model Parameter Efficiency.** Table 3 compares the per-user parameter count per layer across methods. A standard OPPU design requires over 5M parameters per user per layer, since it places independent adapters on all *seven weight matrices* in each transformer block: four in the attention module ($W_q, W_k, W_v, W_o$) and three in the feed-forward module ($W_{\text{up}}, W_{\text{gate}}, W_{\text{down}}$). This heavy footprint makes OPPU impractical to scale across large user populations.

By contrast, our methods dramatically reduce this overhead. The pure SVD ($C_u \approx P_u Q_u$) variant compresses the personalization into a compact shared subspace, achieving a $99.7\%$ reduction in parameter size. APlaud introduces lightweight user-specific corrections and a rank-one residual, requiring only about $2\%$ of the OPPU footprint, while Aplaud+ further factorizes the corrections to reduce usage to under $2\%$. These results highlight the strong parameter efficiency of our framework, which balances compactness with sufficient expressive capacity to yield substantial performance improvements.

Finally, we note that the ablation study and additional experiments can be found in the Appendix.

## 5 CONCLUSION

In this work, we introduced **APlaud**, a scalable and lightweight framework for personalizing large language models (LLMs) at the individual user level in the context of survey response prediction. APlaud leverages a shared low-rank subspace obtained through global LoRA fine-tuning, while enabling user-specific adaptation via a nested low-rank correction and an optional rank-one residual. This design achieves strong personalization with minimal per-user parameter overhead.

Our approach addresses key challenges in user modeling—such as data sparsity and scalability – while consistently outperforming existing LoRA-based personalized methods in both accuracy and parameter efficiency. These results highlight the promise of APlaud for simulating individualized behavior in large-scale settings, offering a principled bridge between global adaptation and fine-grained user representation.

In future work, we plan to extend APlaud to a broader range of personalization tasks, including recommendation systems, writing assistants, and other applications where lightweight and expressive user modeling is critical. We also plan to integrate it with the quantization approach to further reduce parameter space.

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

Table 4: Dataset statistics across six benchmarks. "# Qs" denotes the number of questions, and "Avg Q length" is measured in tokens.

|  | G&L | TS | F&R | EI | GSS | LAMP Movie Tagging |
|---|---|---|---|---|---|---|
| # Users | top 200 | top 200 | top 200 | top 200 | top 200 | top 100 |
| # Qs | 6104 | 7582 | 9036 | 9596 | 6161 | 8860 |
| Avg Q length | 1343.2 | 1678.1 | 1384.5 | 1597.7 | 1195.6 | 572.3 |

## A    EXPERIMENT SETTING

### A.1    ENVIRONMENTAL SETTING

All experiments were conducted on a single cluster node equipped with a Dell PowerEdge C6620 and NVIDIA H100 GPUs with 94 GB of memory.

### A.2    MORE DETAILS ON DATA STATISTICS

In this sections, we first provide all six data statistics we used in this paper in Table 4

We also disclose more details on the missing/incompleteness data statistics and analysis: Across the four ATP waves used in our study, the total number of survey items varies by design (W36: 139 questions, W42: 129, W50: 127, W54: 115). Consistent with ATP's rotating-module structure, raw item-level missingness ranges from 28% to 62%, reflecting that different sub-samples receive different topical modules rather than indicating data quality issues. After applying our quality-control filter (retaining respondents with at least 10 valid answers), missingness decreases in every wave (e.g., W36: 62.1% → 56.1%; W42: 41.4% → 39.5%; W50: 54.8% → 34.4%; W54: 28.4% → 24.4%).

### A.3    TRAINING DETAILS

For a fair comparison, all experiments were trained for 5 epochs. In the Table1, we set the LoRA rank and dimension of $C_u$ to be 64. Subsequently, we selected the top 16 SVD dimension as the starting state for the second training phase, and set the residual module $A_u$ and $B_u$ to rank 1, respectively. Since the initialization of $m$ significantly influence the final result, we tuned its initialization from $\{50.0, 30.0, 20.0, 10.0, 5.0, 2.0, 1.0, 0.1, 0.01, 0.01\}$ and reported the best result. These choices are based on our ablation study, where we explored the LoRA rank from $\{8, 16, 32, 64, 128\}$, SVD dimension from $\{4, 8, 16, 32\}$, training epochs from $\{5, 7, 9, 10\}$ and residual term rank from $\{1, 2, 4, 8\}$. We found that reducing the rank of the pretrained LoRA may boosted performance, while changes in other hyperparameters had less impact.

## B    ABLATION STUDY

In this section, we systematically explore the effect of 4 hyperparameters: LoRA Rank, SVD dim (i.e., rank of $P_u$ and $Q_u$), number of training epochs, and residual dimension (i.e., rank of $A_u$ and $B_u$) on our framework's performance. This analysis helps identify the optimal range for each setting and provides insight into the robustness of our approach. We conduct ablation study on our textbfAplaud+ model. The results of our ablation study are presented in Table 5 and Fig 3. Specifically, in Table 3 (a), we fix the training epoch at 5 and set the residual dimension to 1. Given that the SVD dimension must remain smaller than $C_u$ (i.e., the LoRA rank), we experiment with the following (LoRA rank, SVD dim) pairs: (128, 16), (64, 16), (32, 16), (16, 8), and (8, 4). In Table 5 (b), we set LoRA rank to be 64, SVD dim 16, residual dimension to be 1 and experiment on training epochs from $\{5, 7, 9, 10\}$. In Table 5 (c), we set LoRA rank to be 64, training epoch to be 5, residual dimension to be 1 and experiment on SVD dimension from $\{32, 16, 8, 4\}$. In Table 5 (d), we set LoRA rank to be 64, training epoch to be 5, SVD dimension to be 16 and experiment on different residual dimension from $\{1, 2, 4, 8\}$.

Table 5: Ablation studies across four key hyperparameters. All reported values are test accuracy. G&L = Gender & Law, TS = Twitter Stance, F&R = Finance & Risk, EI = Emotional Intensity, GSS = General Social Survey.

**(a) LoRA rank**

| Rank | G&L | TS | F&R | EI | GSS |
|---|---|---|---|---|---|
| 128 | 0.6715 | 0.7730 | 0.6278 | 0.5000 | 0.4463 |
| 64 | 0.6876 | 0.7862 | 0.6537 | 0.4992 | 0.4605 |
| 32 | 0.6989 | 0.7781 | 0.6667 | 0.4950 | 0.4647 |
| 16 | 0.6940 | 0.7801 | 0.6550 | 0.4983 | 0.4944 |
| 8 | 0.6957 | 0.7649 | 0.6368 | 0.4883 | 0.5071 |

**(b) Training epoch**

| Epoch | G&L | TS | F&R | EI | GSS |
|---|---|---|---|---|---|
| 5 | 0.6876 | 0.7862 | 0.6537 | 0.4992 | 0.4605 |
| 7 | 0.6924 | 0.7781 | 0.6459 | 0.5042 | 0.4449 |
| 9 | 0.6924 | 0.7730 | 0.6515 | 0.5025 | 0.4322 |
| 10 | 0.6957 | 0.7690 | 0.6433 | 0.5042 | 0.4364 |

**(c) SVD dimension**

| SVD dim | G&L | TS | F&R | EI | GSS |
|---|---|---|---|---|---|
| 32 | 0.6860 | 0.7822 | 0.6537 | 0.4933 | 0.4576 |
| 16 | 0.6876 | 0.7862 | 0.6537 | 0.4992 | 0.4605 |
| 8 | 0.6876 | 0.7882 | 0.6459 | 0.5025 | 0.4590 |
| 4 | 0.6795 | 0.7893 | 0.6472 | 0.4950 | 0.4590 |

**(d) Residual dimension**

| Res dim | G&L | TS | F&R | EI | GSS |
|---|---|---|---|---|---|
| 1 | 0.6876 | 0.7862 | 0.6537 | 0.4992 | 0.4605 |
| 2 | 0.6892 | 0.7852 | 0.6511 | 0.4967 | 0.4633 |
| 4 | 0.6892 | 0.7852 | 0.6519 | 0.4967 | 0.4576 |
| 8 | 0.6876 | 0.7761 | 0.6329 | 0.4975 | 0.4449 |

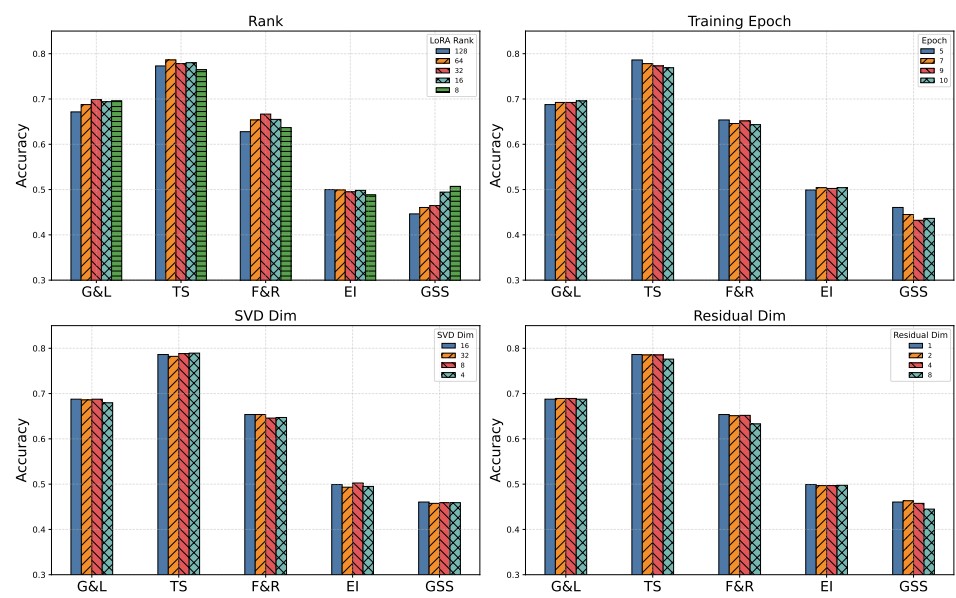

Figure 3: Ablation study on different hyperparameters.

## B.1 LoRA Rank

We observe that the impact of the pretrained LoRA rank on model performance is not consistent across datasets. For example, while accuracy on G&L and F&R improves as the rank decreases, the performance on TS and EI slightly drops when the rank is reduced to 8. This inconsistency suggests that the optimal LoRA rank may vary depending on the dataset characteristics. To ensure a fair comparison under a uniform setting, we additionally conduct experiments using pretrained LoRA with a fixed rank of 8 for all models. The results, summarized in Table 6, show that our framework continues to outperform all baselines under this constrained low-rank setting.

## B.2 Training Epoch

We evaluate the effect of training epochs by varying the number of fine-tuning epochs from 5 to 10. Our observations indicate that extending training beyond 5 epochs does not consistently improve performance. For example, on TS, accuracy decreased from 0.7862 (epoch 5) to 0.7690 (epoch 10), and on F&R, it dropped from 0.6537 to 0.6433. This suggests that the model begins to overfit the limited personalized data after a few epochs. Overall, 5 epochs already yield strong performance, with

additional training offering diminishing or even negative returns, thus emphasizing the importance of early stopping or lightweight adaptation.

### B.3 SVD DIMENSION

To assess the role of the low-rank SVD projection dimension, we compared the SVD dimension (rank of $P_u$ and $Q_u$) of 4, 8, 16, and 32. The best overall performance was achieved using 16 dimensions, which served as our default setting. While increasing the SVD dimension to 32 maintained similar performance on F&R and G&L, it slightly degraded on EI and GSS. Conversely, reducing the dimension to 4 caused a more noticeable drop in performance, particularly on G&L (0.6795). These trends indicate that although the method is robust to reasonable SVD compression, extremely low dimensions may under-represent user-specific variation.

### B.4 RESIDUAL DIMENSION

We varied the dimension of the user-specific residual component (residual dimension, the rank of $A_u$ and $B_u$) from 1 to 8 to assess its effect. Performance remained relatively stable when increasing from 1 to 4 dimensions, showing minor fluctuations across datasets. For example, on TS, performance was nearly unchanged between 1 and 4 dimensions (0.7862 vs. 0.7852), suggesting that even a very low-rank residual term is sufficient to capture key personalization signals. However, setting the residual dimension to 8 lead to a performance drop on several datasets (e.g., G&L and F&R), likely due to overfitting. This result highlights the effectiveness of an extreme small residual dimension.

## C ADDITIONAL EXPERIMENTAL RESULTS

### C.1 RANK-8 PRETRAINED LORA BASED EXPERIMENTAL RESULT

As ablation studies indicated that rank of pretrained LoRA may influence our three-phase training performance across different datasets, for a fair and more comprehensive comparison, we also report experimental results (including baseline results) based on rank-8 pretrained LoRA in Table 6 for better comparison. The experiment setting is as follow: LoRA rank is set to be 8, training epoch is set to be 5, SVD dimension to be 4, rank of residual term ($A_u$ and $B_u$) to be 1, we tune the initlization of $m$ from $\{50.0, 30.0, 20.0, 10.0, 5.0, 2.0, 1.0, 0.1, 0.01, 0.01\}$ and reported the best result. Compared with Table 1, the following conclusions can be drawn:

- While all models are affected by the choice of LoRA rank, APlaud tends to benefit more from lower ranks compared to baseline methods, which sometimes exhibit marginal or negative effects, particularly on datasets like GSS.

- Our APlaud framework outperforms baseline models across all datasets, with the largest improvements reaching up to 12.8% in accuracy and 6.17% in macro-F1.

- These results further highlight the robustness of our model, as it consistently outperforms baselines in personalization tasks regardless of the pre-trained LoRA setup, and does so with substantially fewer parameters.

Table 6: Performance comparison across different datasets based on pretrained rank-8 LoRA. Bold numbers indicate the best results within each dataset. Dataset names: G&L = ATP Gender & Leadership, TS = ATP Trust in Science, F&R = ATP Family and Relationships, EI = ATP Economic Inequality, GSS = General Social Survey.

| Method | G&L | | TS | | F&R | | EI | | GSS | |
|--------|-----|-----|-----|-----|-----|-----|-----|-----|-----|-----|
| | ACC | F1 | ACC | F1 | ACC | F1 | ACC | F1 | ACC | F1 |
| LoRA | 0.6312 | 0.6209 | 0.7123 | 0.4216 | 0.5681 | 0.2655 | 0.4117 | 0.3219 | 0.4901 | 0.3165 |
| OPPU | 0.6828 | 0.6761 | 0.7822 | 0.7004 | 0.6174 | 0.3569 | 0.4950 | 0.4103 | 0.5014 | 0.3349 |
| Cu | 0.6908 | 0.6824 | 0.7903 | 0.6598 | 0.5863 | 0.3100 | 0.4925 | 0.3982 | 0.5042 | 0.3269 |
| Aplaud+ | **0.7005** | **0.6921** | **0.7994** | **0.7436** | **0.6498** | **0.4028** | **0.5008** | **0.4161** | 0.5240 | 0.3484 |

Table 7: Performance comparison across different datasets with LLaMA-2 backbone with profile being removed from prompts. Dataset names: G&L = ATP Gender & Leadership, TS = ATP Trust in Science, F&R = ATP Family and Relationships, EI = ATP Economic Inequality, GSS = General Social Survey.

| Method | G&L | | TS | | F&R | | EI | | GSS | |
|--------|-----|-----|-----|-----|-----|-----|-----|-----|-----|-----|
| | ACC | Macro-F1 | ACC | Macro-F1 | ACC | Macro-F1 | ACC | Macro-F1 | ACC | Macro-F1 |
| LoRA | 0.4576 | 0.3106 | 0.6454 | 0.3249 | 0.5486 | 0.2598 | 0.4725 | 0.3028 | 0.3583 | 0.2139 |
| OPPU | 0.6271 | 0.6030 | 0.7528 | 0.6115 | 0.5970 | 0.3462 | 0.4942 | **0.4005** | 0.3686 | 0.2322 |
| Cu | 0.6151 | 0.5798 | 0.6991 | 0.5838 | 0.5681 | 0.2975 | 0.4942 | 0.3602 | 0.3743 | 0.2545 |
| Aplaud | **0.6457** | **0.6225** | 0.7599 | 0.6165 | 0.6148 | 0.3498 | 0.4942 | 0.3614 | 0.3743 | 0.2604 |
| Aplaud+ | 0.6329 | 0.6079 | **0.7639** | **0.6231** | **0.6200** | **0.3599** | **0.5083** | 0.3918 | **0.3757** | **0.2617** |

Table 8: Generation Task Performance on LaMP News Headline Dataset.

| Method | R-1 | R-L |
|--------|-----|-----|
| GPT5-profile | 0.1312 | 0.1177 |
| OPPU | 0.1987 | 0.1832 |
| Cu | 0.1987 | 0.1825 |
| Aplaud | 0.2003 | 0.1838 |
| Aplaud+ | 0.1987 | 0.1830 |

## C.2 Performance Result without Profile Input

Our user profile is constructed from survey-provided demographic metadata and the 10 survey questions most relevant to user characterization. These questions are selected from the user's answered items and capture personal attitudes, preferences, and values. To avoid information leakage, we remove these 10 profile-related questions before forming the train/validation/test split. See D.3 on the data and prompt we use for profile generation.

To verify the importance of profile information and to evaluate each model's ability to learn user preferences without explicit profile signals, we conduct an additional ablation in which all models are trained and evaluated without any profile input. As shown in Table 7, removing profile information leads to a consistent performance drop across all methods, demonstrating that user profiles provide valuable preference cues. Nevertheless, Aplaud and Aplaud+ remain highly competitive and often surpass OPPU across multiple datasets, despite using significantly fewer per-user parameters. These results indicate that our lightweight modules can effectively recover stable preference patterns even in the absence of explicit profile features, highlighting the robustness and parameter efficiency of our approach.

## C.3 Additional Evaluation on Generation Task

We further evaluate our approach on the LAMP News Headline generation task to assess whether the proposed personalization mechanism also benefits a text generation setting. As shown in Table 8, our Aplaud and Aplaud+ models achieve performance comparable to the strong OPPU baseline while using only about 1% of its per-user parameters.

## C.4 Additional Result where Stage 1 does not use train/test user

To ensure that Stage 1 pretraining does not unintentionally encode information about users who later appear in personalization, we conduct an additional experiment in which Stage 1 is trained on a 20% subsample of users that does not overlap with any train/test users. This setup guarantees that none of the evaluation users contribute to Stage 1 and thus eliminates any possibility of user-level information leaking across stages. As shown in Table 9, the performance of Aplaud and Aplaud+ remains virtually unchanged, confirming that Stage 1 captures only *general task knowledge*, while Stage 2 is solely responsible for learning user-specific preferences. This further demonstrates that task-level learning and personalization are cleanly disentangled in our framework.

Table 9: Performance comparison across different datasets with LLaMA-2 backbone where users in Stage 1 pretraining does not overlap with users in Stage 2

| Method | G&L | | TS | | F&R | | EI | | GSS | |
|--------|-----|-----|-----|-----|-----|-----|-----|-----|-----|-----|
| | ACC | Macro-F1 | ACC | Macro-F1 | ACC | Macro-F1 | ACC | Macro-F1 | ACC | Macro-F1 |
| LoRA | 0.5749 | 0.5344 | 0.6667 | 0.3736 | 0.4929 | 0.2416 | 0.4642 | 0.3687 | 0.3136 | 0.1707 |
| OPPU | 0.6667 | 0.6529 | 0.7305 | 0.5995 | 0.6135 | 0.3471 | 0.5042 | 0.3951 | 0.3771 | 0.2909 |
| Cu | 0.6441 | 0.6425 | 0.7305 | 0.5987 | 0.6196 | 0.3440 | 0.5208 | 0.3955 | 0.3775 | 0.2943 |
| Aplaud | 0.6506 | 0.6506 | 0.7599 | 0.6335 | 0.6265 | 0.3695 | 0.5167 | 0.3927 | **0.4124** | **0.3259** |
| Aplaud+ | **0.6860** | **0.6717** | **0.7639** | **0.6401** | **0.6291** | **0.3706** | **0.5208** | **0.3981** | 0.3969 | 0.3147 |

Table 10: Pratical Running Time and Memory Usage

| Method | SVD Time | Time | GPU Mem | # Per User Param |
|--------|----------|------|---------|------------------|
| LoRA | – | 29 min | 49.18 GB | 100% |
| OPPU | – | 72 min | 53.96 GB | 100% |
| APlaud+:Stage1 | 17 min | 38 min | 67.09 GB | 0.54% |
| APlaud+:Stage2(a) | – | 33 min | 62.04 GB | 0.27% |
| APlaud+:Stage2(b) | 46 s | 34 min | 61.12 GB | 1.84% |
| APlaud+:Stage2(c) | 18 min | 105 min | 67.09 GB | 1.84% |

## C.5 PRATICAL RUNNING TIME AND MEMORY USAGE

In this section, we show practical running time and memory in table 10

## C.6 SENSITIVITY ANALYSIS

We conducted sensitivity analyses to assess the robustness of APlaud to imperfect initializations of the shared subspace and to examine potential error propagation during subsequent training stages.

To simulate noisy conditions, we added Gaussian noise $\epsilon \cdot \mathcal{N}(0, 1)$ to the global LoRA matrices $A$ and $B$ prior to performing SVD:

$$(U, \Sigma, V) = \text{SVD}(AB + \epsilon \cdot \mathcal{N}(0, 1)).$$

This perturbation introduces stochasticity into the shared subspace and allows us to observe the impact of initialization noise on downstream personalization.

We specifically use $\epsilon = 10^{-3}$ and $10^{-4}$, which introduce non-trivial yet controlled noise magnitudes. These values are chosen to reflect realistic perturbations relative to the typical scale of LoRA updates, where $AB$ often has entries on the order of $10^{-2}$ or smaller.

As shown in our experimental results (Table 11), APlaud demonstrates strong robustness to such perturbations, consistently exhibiting low performance variance across runs. This suggests that APlaud does not rely heavily on precise early-stage decompositions and can generalize effectively even in the presence of moderate noise during shared initialization.

## C.7 SHARED SUBAPACE SIMILARITY ANALYSIS

In this section, we analyze the similarity of different shared subpace $U$ and $V$ across different topics. Specifically, we computed average (across different modules within the same layer) *Centered Kernel Alignment* (CKA) (46) similarities of the SVD components $U$ and $V$ between two different topics (G&L vs. TS) as shown in Fig 4: (i) $V$ is relatively stable across topics, while (ii) $U$ varies more. This confirms that $U$ and $V$ serve different purposes: $V$ acts as a coordinate generator which could be relevant across different topics, whereas $U$ serves as different semantic subspaces built on top of $V$. Despite this, both remain well-aligned across topics, supporting our shared-subspace design.

We would like to mention again that our method does not assume that a single pair of $(U, V)$ must generalize across all tasks or topics. In practice, the model learns different $(U, V)$ for different tasks/topics, as also supported by our CKA analysis. The personalized parameters $(C_u, \alpha_u, \beta_u)$ are

Table 11: Robustness study under random noise injection. We report mean $\pm$ std for ACC and Macro-F1.

| Method | ACC | Macro-F1 |
|---|---|---|
| Cu | $0.6626 \pm 0.0031$ | $0.6455 \pm 0.0070$ |
| PuQu | $0.6618 \pm 0.0039$ | $0.6411 \pm 0.0038$ |
| Aplaud+ | $\mathbf{0.6856 \pm 0.0051}$ | $\mathbf{0.6686 \pm 0.0062}$ |
| Cu | $0.6642 \pm 0.0009$ | $0.6435 \pm 0.0011$ |
| PuQu | $0.6626 \pm 0.0059$ | $0.6424 \pm 0.0058$ |
| Aplaud+ | $\mathbf{0.6852 \pm 0.0016}$ | $\mathbf{0.6677 \pm 0.0020}$ |
| Cu | $0.6611 \pm 0.0060$ | $0.6401 \pm 0.0058$ |
| PuQu | $0.6654 \pm 0.0038$ | $0.6432 \pm 0.0048$ |
| Aplaud+ | $\mathbf{0.6755 \pm 0.0093}$ | $\mathbf{0.6593 \pm 0.0050}$ |

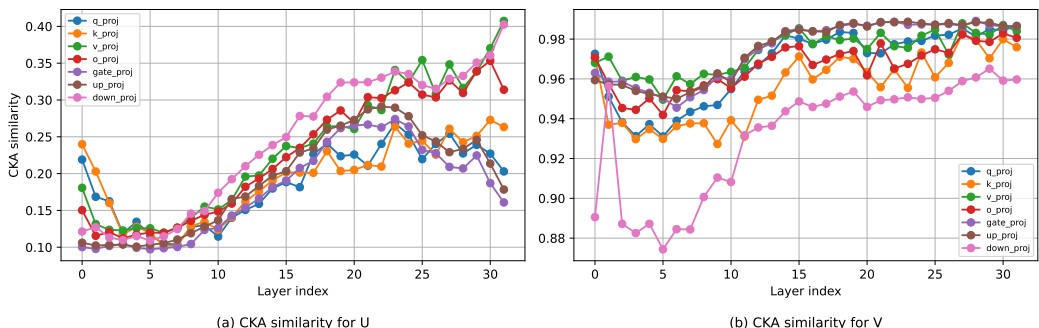

(a) CKA similarity for U

(b) CKA similarity for V

Figure 4: Average CKA similarity of U, V under different layers across different topic

then learned in Stage 2 within the subspace defined by that task/topic-specific $(U, V)$. This design ensures that personalization is performed inside an already aligned subspace, without requiring cross-topic invariance of $(U, V)$.

Regarding drift across waves and time, our current datasets mix users from different time points within each topic, so we cannot cleanly isolate purely temporal drift in this paper. We view a more fine-grained, time-indexed analysis as important future work. However, in real survey practice, questions within the same topic typically evolve slowly and remain within a relatively narrow semantic range. Combined with the high CKA stability of $V$ that we observe, this gives us good reason to believe that subspace drift across waves and time within a topic is gradual rather than catastrophic in the survey setting we target.

## C.8 SIGNIFICANCE

We conducted 5 runs with different random seeds on two representative datasets with Llama2 backbone and compared against SOTA. The results, reported in 12, show that APlaud consistently and robustly outperforms baselines. Due to time constraints, we could not repeat all settings, but we will include the full repeated results in the camera-ready version.

## C.9 RELATIVE OF IMPROVEMENT PERFORMANCE OVER DIFFERENT BASELINES

In this section, we present the Relative of Improvement (ROI) of our method (we report results of Aplaud+ as a representative here) over all other baselines in Table 13 and Table 14.

Compared with non-personalized PEFT methods (LoRA, PiSSA, MiLoRA, AdaLoRA, QLoRA), Aplaud+ provides clear and stable gains. For example, on Llama2-7B it improves Macro-F1 by +27.7% on TS and +10.6% on F&R over LoRA, and on Mistral-7B it further increases TS Macro-F1

Table 12: Siginificance performance comparison across different methods (mean ± std over 5 runs).

| Method | G&L | | LAMP Movie Tagging | |
|---|---|---|---|---|
| | ACC | F1 | ACC | F1 |
| *Non-Personalized* | | | | |
| LoRA | 0.6402 ± 0.0052 | 0.6174 ± 0.0083 | 0.6220 ± 0.0082 | 0.5037 ± 0.0074 |
| QLoRA | 0.6315 ± 0.0149 | 0.6192 ± 0.0175 | 0.6257 ± 0.0097 | 0.5045 ± 0.0247 |
| PiSSA | 0.6588 ± 0.0116 | 0.6501 ± 0.0127 | 0.6274 ± 0.0103 | 0.5172 ± 0.0074 |
| MiLoRA | 0.6620 ± 0.0079 | 0.6581 ± 0.0130 | 0.6297 ± 0.0068 | 0.5298 ± 0.0063 |
| AdaLoRA | 0.6637 ± 0.0113 | 0.6503 ± 0.0148 | 0.6201 ± 0.0139 | 0.5102 ± 0.0083 |
| *Personalized* | | | | |
| OPPU | 0.6602 ± 0.0094 | 0.6423 ± 0.0187 | 0.6356 ± 0.0106 | 0.5174 ± 0.0115 |
| Cu | 0.6627 ± 0.0071 | 0.6507 ± 0.0263 | 0.6410 ± 0.0103 | 0.5289 ± 0.0137 |
| Aplaud+ | **0.6785 ± 0.0040** | **0.6657 ± 0.0082** | **0.6602 ± 0.0079** | **0.5512 ± 0.0091** |

Table 13: Relative of Improvements of Aplaud+ over different baselines across datasets with llama2-7B backbone.

| Method | G&L | | TS | | F&R | | EI | | GSS | | LAMP MV | | Avg ACC Gain | Avg F1 Gain |
|---|---|---|---|---|---|---|---|---|---|---|---|---|---|---|
| | ACC | Macro-F1 | ACC | Macro-F1 | ACC | Macro-F1 | ACC | Macro-F1 | ACC | Macro-F1 | ACC | Macro-F1 | | |
| **Non-Personalized** | | | | | | | | | | | | | | |
| LoRA | +6.8% | +8.2% | +13.1% | +27.7% | +10.6% | +10.6% | +12.3% | +26.4% | +4.9% | +9.7% | +6.1% | +8.9% | +8.9% | +15.2% |
| PiSSA | +8.4% | +8.1% | +8.0% | +21.4% | +16.6% | +8.1% | +8.4% | +12.3% | +11.5% | +16.5% | +6.3% | +4.7% | +9.9% | +11.9% |
| MiLoRA | +1.7% | +0.1% | +7.7% | +18.3% | +15.0% | +5.7% | +12.2% | +18.7% | +3.5% | +1.4% | +4.5% | +3.5% | +7.4% | +8.0% |
| AdaLoRA | +1.9% | -0.5% | +2.5% | +14.4% | +14.7% | +2.8% | +9.9% | +13.3% | +1.6% | -4.1% | +7.3% | +9.3% | +6.3% | +5.9% |
| QLoRA | +3.2% | +2.6% | +6.8% | +28.4% | +18.0% | +8.6% | +10.7% | +26.8% | +6.2% | +7.4% | +4.6% | +5.3% | +8.2% | +13.2% |
| **Personalized** | | | | | | | | | | | | | | |
| GPT5-profile | +26.6% | +24.4% | +29.2% | +97.9% | +28.8% | +28.7% | +13.5% | +19.7% | -15.4% | +5.8% | +20.4% | +22.7% | +17.2% | +33.2% |
| GPT5-RAG | +7.1% | +5.3% | +13.9% | +10.6% | +3.4% | +4.6% | -0.6% | +1.5% | -22.3% | -22.8% | – | – | +0.3% | -0.1% |
| OPPU | +2.7% | +1.3% | +5.6% | +8.7% | +4.7% | +2.5% | +3.5% | +2.2% | +7.2% | +5.1% | +4.1% | +7.4% | +4.6% | +4.5% |

by +51.1% over AdaLoRA. These results indicate that generic finetuning cannot capture user-specific heterogeneity, while Aplaud+ effectively personalizes model behavior.

Compared with retrieval-based personalization (GPT5-profile / GPT5-RAG), Aplaud+ delivers significantly higher and more robust performance. On Llama2-7B, it surpasses GPT5-profile by +17.2% ACC and +33.2% Macro-F1 on average. Aplaud avoids dependence on prompt context quality and instead encodes stable user preferences in parameters.

Compared with OPPU, a strong personalized baseline, Aplaud still achieves consistent improvements. On Llama2-7B it yields +4.6% ACC and +4.5% Macro-F1 gains on average, and on Mistral-7B the gains reach +2.5% ACC and +10.2% Macro-F1. This demonstrates the superiority of our meticulously designed personalized modules to capture user preferences.

### C.10 WASSERSTEIN DISTANCE RESULT

As an addition to Table 1, Table 15 presents the results on approximating human responses for Pew Research Center surveys and the General Social Survey under the same setting. We report Accuracy (ACC), F1 Score (F1), and Wasserstein Distance (WD), with WD measuring the average distributional distance between real human subjects and simulated virtual subjects across test survey questions. A lower WD indicates better distributional alignment. Across all datasets, the **Aplaud+** method consistently achieves the **lowest Wasserstein Distance (WD)**, indicating superior alignment between the distributions of human and simulated responses. For instance, on the **GSS** dataset, the WD achieved by Aplaud+ is **0.1949**, outperforming both the **LoRA** baseline (0.2782) and other personalized approaches. This trend holds across various survey domains:

- In **Gender and Leadership (G&L)**, our method achieves a WD of **0.0097**, which is substantially lower than LoRA (0.1111) and OPPU (0.0725).

- In **Trust in Science (TS)**, Cu records a WD of **0.0203**, outperforming LoRA (0.0294) and OPPU (0.0324).

- For **Friendship and Relationships (F&R)**, the WD drops to **0.1362**, compared to **0.4462** with LoRA and **0.2127** with OPPU.

Table 14: Relative of Improvements of Aplaud+ over different baselines with Mistral-7B backbone.

| Method | G&L | | TS | | F&R | | EI | | GSS | | LAMP MV | | Avg ACC Gain | Avg F1 Gain |
|---|---|---|---|---|---|---|---|---|---|---|---|---|---|---|
| | ACC | Macro-F1 | ACC | Macro-F1 | ACC | Macro-F1 | ACC | Macro-F1 | ACC | Macro-F1 | ACC | Macro-F1 | | |
| **Non-Personalized** | | | | | | | | | | | | | | |
| LoRA | +4.9% | +6.3% | +11.2% | +72.1% | +15.1% | +84.2% | +6.0% | +14.9% | +6.2% | +31.8% | +6.2% | +3.4% | +8.3% | +35.5% |
| PiSSA | +5.4% | +4.3% | +6.6% | +39.4% | +20.9% | +83.0% | +10.3% | +23.8% | +12.0% | +22.2% | +4.1% | +1.8% | +9.9% | +29.1% |
| MiLoRA | +4.4% | +3.7% | +5.6% | +42.9% | +18.0% | +69.8% | +7.7% | +17.0% | +1.6% | +9.9% | +3.8% | +3.8% | +6.9% | +24.5% |
| AdaLoRA | +7.0% | +6.7% | +11.2% | +51.1% | +8.9% | +19.6% | +6.0% | +23.9% | -18.5% | -3.2% | +6.7% | -2.7% | +3.5% | +15.8% |
| QLoRA | +5.7% | +5.9% | +9.5% | +40.5% | +20.5% | +75.7% | +6.2% | +16.7% | +3.8% | +19.3% | +0.5% | -5.7% | +7.7% | +25.4% |
| **Personalized** | | | | | | | | | | | | | | |
| GPT5-profile | +27.4% | +26.3% | +27.4% | +108.9% | +31.9% | +51.8% | +9.3% | +10.6% | -1.7% | +34.6% | +29.2% | +15.4% | +20.6% | +41.3% |
| GPT5-RAG | +7.8% | +6.9% | +12.3% | +16.8% | +5.9% | +23.3% | -4.2% | -6.3% | -9.8% | -1.7% | – | – | +2.4% | +7.8% |
| OPPU | +2.1% | +2.8% | +0.1% | +3.5% | +2.7% | +19.6% | +1.3% | -1.9% | +6.5% | +19.0% | +2.4% | +18.3% | +2.5% | +10.2% |

Table 15: WD Performance across all survey datasets.

| Method | G&L | | | TS | | | F&R | | | EI | | | GSS | | |
|---|---|---|---|---|---|---|---|---|---|---|---|---|---|---|---|
| | ACC | F1 | WD | ACC | F1 | WD | ACC | F1 | WD | ACC | F1 | WD | ACC | F1 | WD |
| LoRA | 0.6554 | 0.6342 | 0.1111 | 0.7072 | 0.4186 | 0.0294 | 0.5681 | 0.2372 | 0.4462 | 0.4708 | 0.3356 | 0.3350 | 0.4336 | 0.2624 | 0.2782 |
| OPPU | 0.6731 | 0.6560 | 0.0725 | 0.7852 | 0.6962 | 0.0324 | 0.6368 | 0.3654 | 0.2127 | 0.4925 | **0.3936** | **0.0933** | 0.4322 | 0.2906 | 0.1977 |
| Cu | 0.6828 | 0.6691 | **0.0097** | 0.7781 | 0.6338 | **0.0203** | 0.5746 | 0.2796 | 0.4047 | 0.5008 | 0.3868 | 0.1625 | 0.4548 | 0.2917 | 0.2274 |
| Aplaud+ | **0.6876** | **0.6742** | **0.0097** | **0.7862** | **0.7204** | 0.0324 | **0.6537** | **0.4369** | **0.1362** | 0.4992 | 0.3859 | 0.1558 | **0.4605** | **0.3459** | **0.1949** |

- In **Economic Inequality (EI)**, our model achieves a WD of **0.1558**, improving over LoRA (0.3350) and slightly increasing to OPPU (0.0933).

- In **Economic Inequality (EI)**, our model achieves a WD of **0.1558**, improving over LoRA (0.3350) and slightly increasing to OPPU (0.0933).

The results underscore the effectiveness of structure-aware personalization, particularly the bias-corrected matrix factorization **Aplaud+**, in accurately capturing subtle, user-specific behavioral patterns across diverse survey domains.

### C.11 SUBGROUP COMPARISONS ACROSS DEMOGRAPHIC VARIABLES

To assess how well each personalized model generalizes across diverse population subgroups, we conduct stratified evaluations along three key demographic dimensions: geographic region (CREGION), sex (SEX), and political affiliation (POLPARTY). This analysis spans four waves of the Pew American Trends Panel (ATP)—Waves 36, 42, 50, and 54—as well as the 2016–2020 General Social Survey (GSS) Panel.

For each dataset, we report model performance within each subgroup using two metrics: classification accuracy (ACC) and Wasserstein Distance (WD). We highlight the best-performing results in each subgroup to assess the consistency, fairness, and personalization quality of different adaptation methods.

**Subgroup definitions:**

- **CREGION** $\in$ {Northeast, Midwest, South, West}
- **SEX** $\in$ {Male, Female}
- **POLPARTY** $\in$ {Republican, Democrat, Independent, Other}

#### C.11.1 ATP WAVE 36

**Subgroup Performance in ATP Wave 36 by Region (CREGION).** As shown in Table 16, **Aplaud+** yields the highest accuracy in the West (0.7671) and consistently performs well across other regions. In contrast, OPPU achieved the lowest Wasserstein Distance in the Northeast (0.0323), indicating stronger distributional alignment in that specific subgroup. Overall, these results reinforce the strength of structured personalization in APlaud, which combines shared matrix decomposition with user-specific adaptation.

**Subgroup Performance in ATP Wave 36 by Gender (SEX).** As shown in Table 17, **Aplaud+** achieves the highest classification accuracy for both Female (0.6849) and Male (0.6914) respondents.

Table 16: ATP Wave 36: Performance by Region (CREGION).

| Model | Midwest | | Northeast | | South | | West | |
|---|---|---|---|---|---|---|---|---|
| | Acc | WD | Acc | WD | Acc | WD | Acc | WD |
| LoRA | 0.6552 | 0.1437 | 0.5376 | 0.1613 | **0.6683** | 0.0913 | 0.7123 | 0.0685 |
| OPPU | **0.6782** | 0.0920 | 0.5914 | **0.0323** | 0.6394 | 0.0817 | 0.7671 | 0.0753 |
| Cu | 0.6667 | 0.0345 | **0.6559** | 0.1075 | 0.6538 | **0.0529** | 0.7603 | 0.0753 |
| PuQu | 0.6724 | **0.0230** | 0.6452 | 0.0968 | 0.6538 | 0.0577 | 0.7671 | 0.0890 |
| Aplaud+ | **0.6782** | 0.0287 | **0.6559** | 0.1075 | 0.6538 | 0.0577 | **0.7671** | **0.0616** |

In terms of distributional alignment, our method showed significant gains. For Female users, **PuQu** achieved a Wasserstein Distance (WD) of 0.0055, representing a relative reduction of 92.8% compared to LoRA (0.0767), and 90.4% compared to OPPU (0.0575). For Male users, **Aplaud+** yields the lowest WD (0.0469), making a 70.7% improvement over LoRA (0.1602), and 50.0% better than OPPU (0.0938).

Table 17: ATP Wave 36: Performance by Gender (SEX). Best values per column are **bolded**.

| Model | Female | | Male | |
|---|---|---|---|---|
| | Acc | WD | Acc | WD |
| LoRA | 0.6521 | 0.0767 | 0.6602 | 0.1602 |
| OPPU | 0.6685 | 0.0575 | 0.6797 | 0.0938 |
| Cu | 0.6795 | 0.0219 | 0.6875 | 0.0547 |
| PuQu | 0.6795 | **0.0055** | 0.6914 | 0.0547 |
| Aplaud+ | **0.6849** | 0.0164 | **0.6914** | **0.0469** |

**Subgroup Performance in ATP Wave 36 by Political Affiliation (POLPARTY).** Table 18 presents model performance stratified by political affiliation—Democrat, Republican, Independent, and Other—based on ATP Wave 36. **Aplaud+** achieved the highest classification accuracy for both Democrats (0.7095) and Independents (0.6968), while Cu and PuQu also demonstrated consistently strong generalization across subgroups.

In terms of distributional alignment, OPPU achieved the lowest Wasserstein Distance for Democrats (0.0207) and Independents (0.0194), whereas **Aplaud+** performs best for Republicans (0.0733) and users classified as Other (0.1176). These findings suggest that structured personalization approaches, such as APlaud can effectively adapt to diverse political profiles, yielding both accurate and distributionally faithful response simulations.

Table 18: ATP Wave 36: Performance by Political Party (POLPARTY).

| Model | Democrat | | Republican | | Independent | | Other | |
|---|---|---|---|---|---|---|---|---|
| | Acc | WD | Acc | WD | Acc | WD | Acc | WD |
| LoRA | 0.6763 | 0.0830 | 0.6492 | 0.3455 | 0.6258 | 0.1161 | **0.6765** | 0.1471 |
| OPPU | 0.6805 | **0.0207** | **0.6649** | 0.1675 | 0.6839 | **0.0194** | 0.6176 | 0.1471 |
| Cu | 0.7054 | 0.0456 | 0.6545 | 0.0785 | 0.6903 | 0.0258 | 0.6471 | **0.1176** |
| PuQu | 0.7012 | 0.0373 | **0.6649** | 0.0995 | 0.6903 | 0.0258 | 0.6471 | **0.1176** |
| Aplaud+ | **0.7095** | 0.0498 | 0.6597 | **0.0733** | **0.6968** | 0.0258 | 0.6471 | **0.1176** |

C.11.2 ATP WAVE 42

*Topic: Trust in Science*

**Subgroup Performance in ATP Wave 42 by Region (CREGION).** Table 19 presents the subgroup performance across geographic regions. The **Cu** model achieved the highest accuracy in the Midwest

(0.8008) and the lowest Wasserstein Distance (WD) of 0.0456, indicating strong performance in this region. In the Northeast, **PuQu** attains the best accuracy (0.7574) and the lowest WD (0.0221), reflecting excellent distributional alignment. While **Aplaud+** demonstrates the highest accuracy in both the South (0.8232) and West (0.7766), it did not achieve the lowest WD in these regions, suggesting that its distributional alignment was not optimal compared to other models.

Table 19: ATP Wave 42: Performance by Region (CREGION).

| Model | Midwest | | Northeast | | South | | West | |
|---|---|---|---|---|---|---|---|---|
| | Acc | WD | Acc | WD | Acc | WD | Acc | WD |
| LoRA | 0.7178 | 0.0539 | 0.6985 | 0.0368 | 0.7226 | **0.0244** | 0.6844 | 0.0390 |
| OPPU | 0.7759 | 0.0705 | 0.7574 | 0.0294 | 0.8171 | 0.0396 | 0.7695 | 0.0142 |
| Cu | **0.8008** | **0.0456** | 0.7426 | 0.0221 | 0.7866 | 0.0274 | 0.7660 | 0.0177 |
| PuQu | 0.7967 | 0.0498 | **0.7574** | **0.0221** | 0.7835 | 0.0305 | 0.7695 | **0.0142** |
| Aplaud+ | 0.7801 | 0.0664 | 0.7279 | 0.0809 | **0.8232** | 0.0396 | **0.7766** | 0.0213 |

**Subgroup Performance in ATP Wave 42 by Gender (SEX).** Table 20 presents model performance by gender for ATP Wave 42, focusing on Trust in Science. **Aplaud+** achieves the highest accuracy among female respondents (0.7805), whereas OPPU yields the highest accuracy for male respondents (0.7955). In terms of Wasserstein Distance, reflecting distributional alignment, **Cu**, **PuQu**, and **Aplaud+** achieve equally strong alignment (WD = 0.0561) among female respondents. For male respondents, **PuQu** demonstrates the best distributional alignment, achieving the lowest WD (0.0035). These results underscore that structured personalization methods, particularly those incorporating low-rank decomposition and residual correction, effectively enhance prediction accuracy and response alignment across gender subgroups.

Table 20: ATP Wave 42: Performance by Gender (SEX).

| Model | Female | | Male | |
|---|---|---|---|---|
| | Acc | WD | Acc | WD |
| LoRA | 0.7098 | 0.0756 | 0.7054 | 0.0451 |
| OPPU | 0.7707 | 0.0585 | **0.7955** | 0.0139 |
| Cu | 0.7780 | **0.0561** | 0.7782 | 0.0052 |
| PuQu | 0.7780 | **0.0561** | 0.7799 | **0.0035** |
| Aplaud+ | **0.7805** | **0.0561** | 0.7903 | 0.0191 |

**Subgroup Performance in ATP Wave 42 by Political Affiliation (POLPARTY).** Table 21 summarizes performance across political affiliation subgroups for ATP Wave 42, focused on Trust in Science. **Aplaud+** achieves the highest accuracy among Democrats (0.8293), while OPPU yielded the best accuracy for Republicans (0.8143) and Independents (0.7838). **Cu** provided the highest accuracy among respondents identifying as "Other" (0.7828). Regarding Wasserstein Distance (WD), OPPU has the lowest WD among Democrats (0.0585), **Aplaud+** achieved the lowest WD for Republicans (0.0643), **Cu** performed best for Independents (0.0113), and **LoRA** obtains the lowest WD for "Other" affiliations (0.0505).

C.11.3 ATP WAVE 50

*Topic: Family and Relationsihp*

**Subgroup Performance in ATP Wave 50 by Region (CREGION).** Table 22 summarizes model performance across U.S. regions for ATP Wave 50, which focuses on Family and Relationship topics. The **Aplaud+** model achieved the highest accuracy in the Northeast (0.7023), South (0.6128) and West (0.6667) regions. Additionally, **Aplaud+** achieves the lowest Wasserstein Distance (WD) values in four regions, highlighting its superior alignment with real response distributions in these regions. The OPPU model attains the highest accuracy in the Midwest (0.6715), along with competitive WD

Table 21: ATP Wave 42: Performance by Political Party (POLPARTY).

| Model | Democrat | | Republican | | Independent | | Other | |
|---|---|---|---|---|---|---|---|---|
| | Acc | WD | Acc | WD | Acc | WD | Acc | WD |
| LoRA | 0.6976 | 0.0732 | 0.7429 | 0.1286 | 0.7095 | 0.0541 | 0.6869 | **0.0505** |
| OPPU | 0.8244 | **0.0585** | **0.8143** | 0.0714 | **0.7838** | 0.0405 | 0.7273 | 0.0707 |
| Cu | 0.7902 | 0.0829 | 0.7929 | 0.0786 | 0.7658 | **0.0113** | **0.7828** | 0.0556 |
| PuQu | 0.7854 | 0.0780 | 0.8000 | 0.0714 | 0.7725 | 0.0180 | 0.7727 | 0.0657 |
| Aplaud+ | **0.8293** | 0.0732 | 0.8071 | **0.0643** | 0.7748 | 0.0338 | 0.7525 | 0.0657 |

performance. These results demonstrate that structured personalization, particularly through the **Aplaud+** method, significantly enhances both accuracy and distributional fidelity across geographic subpopulations.

Table 22: ATP Wave 50: Performance by Region (CREGION).

| Model | Midwest | | Northeast | | South | | West | |
|---|---|---|---|---|---|---|---|---|
| | Acc | WD | Acc | WD | Acc | WD | Acc | WD |
| LoRA | 0.5839 | 0.4088 | 0.5878 | 0.4504 | 0.6015 | 0.4286 | 0.5105 | 0.4852 |
| OPPU | **0.6715** | 0.2409 | 0.6794 | 0.2137 | 0.5940 | 0.1316 | 0.6414 | 0.2869 |
| Cu | 0.6277 | 0.4015 | 0.6031 | 0.4351 | 0.5677 | 0.3383 | 0.5359 | 0.4641 |
| PuQu | 0.6204 | 0.4088 | 0.6031 | 0.4351 | 0.5639 | 0.3459 | 0.5359 | 0.4641 |
| Aplaud+ | 0.6642 | **0.1898** | **0.7023** | **0.0992** | **0.6128** | **0.0977** | **0.6667** | **0.2025** |

**Subgroup Performance in ATP Wave 50 by Gender (SEX).** As reported in Table 23, **Aplaud+** achieves the best accuracy for both Female (0.6510) and Male (0.6590) subgroups. In addition, it showed the strongest distributional alignment, with WDs of 0.1000 and 0.2069 respectively. This supports the robustness of our personalized decomposition strategy between genders.

Table 23: ATP Wave 50: Performance by Gender (SEX).

| Model | Female | | Male | |
|---|---|---|---|---|
| | Acc | WD | Acc | WD |
| LoRA | 0.5471 | 0.4333 | 0.6092 | 0.4713 |
| OPPU | 0.6392 | 0.1706 | 0.6322 | 0.2950 |
| Cu | 0.5451 | 0.3765 | 0.6322 | 0.4598 |
| PuQu | 0.5431 | 0.3804 | 0.6284 | 0.4636 |
| Aplaud+ | **0.6510** | **0.1000** | **0.6590** | **0.2069** |

**Subgroup Performance in ATP Wave 50 by Political Affiliation (POLPARTY).** Table 24 illustrates the performance across political affiliation subgroups in ATP Wave 50, focusing on family and relationship issues. The **Aplaud+** model demonstrates superior accuracy among Democrats (0.6349) and Republicans (0.7137), as well as competitive performance for Independents (0.5644) and Others (0.6442). Additionally, **Aplaud+** achieves the lowest Wasserstein Distance (WD) for Democrats (0.1905) and Republicans (0.0745). The OPPU method also shows strong performance, especially among Independents, attaining both the highest accuracy (0.5743) and lowest WD (0.0990). These findings emphasize that structured personalization, particularly the **Aplaud+** approach, effectively captures nuanced subgroup differences and improves alignment with real-world response distributions across political affiliations.

C.11.4    ATP Wave 54

*Topic: Economic Inequality*

Table 24: ATP Wave 50: Performance by Political Party (POLPARTY).

| Model | Democrat | | Republican | | Independent | | Other | |
|---|---|---|---|---|---|---|---|---|
| | Acc | WD | Acc | WD | Acc | WD | Acc | WD |
| LoRA | 0.5516 | 0.4405 | 0.6380 | 0.5460 | 0.5686 | 0.4118 | 0.4950 | 0.3861 |
| OPPU | 0.5952 | 0.2460 | 0.6980 | 0.2078 | **0.5743** | **0.0990** | **0.6442** | 0.4356 |
| Cu | 0.5595 | 0.4286 | 0.5922 | 0.3725 | 0.5050 | 0.3168 | 0.6135 | 0.4969 |
| PuQu | 0.5556 | 0.4325 | 0.5922 | 0.3765 | 0.4950 | 0.3267 | 0.6135 | 0.4969 |
| Aplaud+ | **0.6349** | **0.1905** | **0.7137** | **0.0745** | 0.5644 | **0.0990** | **0.6442** | **0.2822** |

**Subgroup Performance in ATP Wave 54 by Region (CREGION).** As shown in Table 25, **Aplaud+** achieves the highest accuracy in the Northeast (0.6176). Additionally, the model OPPU attaining the lowest Wasserstein Distance (WD) in all regions: Midwest (0.1518), Northeast (0.0490), South (0.0889), and West (0.0810).

Table 25: ATP Wave 54: Performance by Region (CREGION).

| Model | Midwest | | Northeast | | South | | West | |
|---|---|---|---|---|---|---|---|---|
| | Acc | WD | Acc | WD | Acc | WD | Acc | WD |
| LoRA | 0.4167 | 0.3690 | 0.5539 | 0.3431 | 0.4578 | 0.3333 | **0.5048** | 0.2762 |
| OPPU | **0.4821** | **0.1518** | 0.5539 | **0.0490** | 0.4822 | **0.0889** | 0.4714 | **0.0810** |
| Cu | 0.4613 | 0.1994 | 0.6078 | 0.1618 | 0.4844 | 0.1667 | 0.4952 | 0.1333 |
| PuQu | 0.4613 | 0.1935 | 0.6029 | 0.1716 | **0.4867** | 0.1733 | 0.5000 | 0.1381 |
| Aplaud+ | 0.4583 | 0.1964 | **0.6176** | 0.1422 | 0.4778 | 0.1578 | 0.4952 | 0.1381 |

**Subgroup Performance in ATP Wave 54 by Gender (SEX).** Table 26 presents the subgroup performance by gender for ATP Wave 54. **Aplaud+** achieves the highest accuracy for both Female (0.5000) and Male (0.4980) respondents. In terms of distributional alignment measured by Wasserstein Distance (WD), OPPU performs best, achieving the lowest WD values for both Female (0.1092) and Male (0.0714) groups. This indicates that while **Aplaud+** was effective in maximizing predictive accuracy, OPPU better captures the nuanced distributional patterns across gender groups.

Table 26: ATP Wave 54: Performance by Gender (SEX).

| Model | Female | | Male | |
|---|---|---|---|---|
| | Acc | WD | Acc | WD |
| LoRA | **0.5000** | 0.2974 | 0.4306 | 0.3869 |
| OPPU | 0.5086 | **0.1092** | 0.4702 | **0.0714** |
| Cu | 0.5043 | 0.1494 | 0.4960 | 0.1806 |
| PuQu | 0.5057 | 0.1566 | 0.4960 | 0.1905 |
| Aplaud+ | **0.5000** | 0.1408 | **0.4980** | 0.1766 |

**Subgroup Performance in ATP Wave 54 by Political Party (POLPARTY).** Table 27 summarizes model performance across political party subgroups for ATP Wave 54. **Aplaud+** achieved the highest accuracy among Democrats (0.5317), while **PuQu** had the best accuracy for Republicans (0.5072) and **Cu** had the best accuracy for Independents (0.5072). **LoRA** performed best in terms of accuracy for the "Other" category (0.4833). Regarding Wasserstein Distance (WD), OPPU showed distributional alignment, yielding the lowest WD values for Democrats (0.0952), Independents (0.0560), and "Other" affiliations (0.2611). For Republicans, **Aplaud+** achieves the lowest WD (0.1307).

C.11.5   GSS PANEL (2016–2020)

*Topic: General Social Trends*
We applied the same stratified evaluation to the General Social Survey panel dataset, using Wave 1a

Table 27: ATP Wave 54: Performance by Political Party (POLPARTY).

| Model | Democrat | | Republican | | Independent | | Other | |
|---|---|---|---|---|---|---|---|---|
| | Acc | WD | Acc | WD | Acc | WD | Acc | WD |
| LoRA | 0.5119 | 0.3492 | 0.3750 | 0.4028 | 0.4626 | 0.3376 | **0.4833** | 0.3444 |
| OPPU | 0.5198 | **0.0952** | 0.4583 | 0.1389 | 0.4957 | **0.0560** | 0.4556 | **0.2611** |
| Cu | 0.5317 | 0.1865 | 0.3889 | 0.2778 | **0.5072** | 0.1394 | 0.4778 | 0.3056 |
| PuQu | 0.5317 | 0.2024 | **0.5072** | 0.1466 | 0.4833 | 0.2778 | 0.3889 | 0.2778 |
| Aplaud+ | **0.5317** | 0.1786 | 0.5057 | **0.1307** | 0.4778 | 0.3000 | 0.3750 | 0.2917 |

variables (2016). The table below summarizes model performance by subgroup. Unlike ATP, GSS includes different varialbes for region.

**Subgroup Performance in GSS by Region (CREGION).**   As shown in Table 28, the **Aplaud+** model (APlaud) achieved the highest accuracy and lowest Wasserstein Distance in both SoNew England and Pacific regions, indicating particularly strong performance in these areas. In contrast, simpler models such as LoRA and OPPU perform best in different regions, with LoRA attaining its highest accuracy in the Middle Atlantic (Acc = 0.4171) and OPPU in East North Central (Acc = 0.3596), while OPPU's lowest WD is observed in the Middle Atlantic (0.2362).

Table 28: GSS: Performance by Sub-Region (CREGION)

| Model | East North Central | | Middle Atlantic | | SoNew England | | Pacific | |
|---|---|---|---|---|---|---|---|---|
| | Acc | WD | Acc | WD | Acc | WD | Acc | WD |
| LoRA | 0.3483 | **0.1910** | **0.4171** | 0.3266 | 0.4340 | 0.3113 | 0.4545 | 0.3455 |
| OPPU | **0.3596** | **0.1910** | 0.4070 | **0.2362** | 0.5094 | 0.2453 | 0.4364 | 0.2424 |
| Cu | 0.3371 | 0.2584 | 0.3970 | 0.2714 | 0.5283 | 0.2264 | 0.4788 | 0.2364 |
| PuQu | 0.3371 | 0.2584 | 0.3970 | 0.2714 | 0.5377 | 0.2170 | 0.4788 | 0.2364 |
| Aplaud+ | 0.3146 | 0.2360 | 0.3970 | **0.2362** | 0.5566 | **0.1887** | **0.5091** | **0.2061** |

**Subgroup Performance in GSS by Gender (SEX).**   Table 29 presents performance by gender for the GSS data. **PuQu** achieved the highest accuracy for females (0.4936), while **Cu** attained the highest accuracy for males (0.4873). In terms of Wasserstein Distance (WD), **PuQu** yields the lowest WD for females (0.1911), and OPPU achieved the lowest WD for males (0.1656).

Table 29: GSS: Performance by Gender (SEX).

| Model | Female | | Male | |
|---|---|---|---|---|
| | Acc | WD | Acc | WD |
| LoRA | 0.4061 | 0.3198 | 0.4682 | 0.2580 |
| OPPU | 0.4162 | 0.2234 | 0.4522 | **0.1656** |
| Cu | 0.4289 | 0.2487 | **0.4873** | 0.2006 |
| PuQu | **0.4936** | **0.1911** | 0.4676 | 0.1877 |
| Aplaud+ | 0.4442 | 0.2081 | 0.4809 | 0.1847 |

**Subgroup Performance in GSS by Political Party (POLPARTY).**   Table 30 presents model performance by political party affiliation. For Democrats, our APlaud approach (**Aplaud+**) achieves the highest accuracy (0.4778). Among Independents and respondents identifying as Other, several models—including OPPU, Cu, PuQu, and Aplaud+—reached the maximum possible accuracy (0.5000). In terms of Wasserstein Distance (WD), both APlaud and OPPU achieve the lowest value (0.1667) for the Other group, while APlaud yielded the lowest WD for Independents (0.2038).

Table 30: GSS: Performance by Political Party (POLPARTY)

| Model | Democrat | | Republican | | Independent | | Other | |
|---|---|---|---|---|---|---|---|---|
| | Acc | WD | Acc | WD | Acc | WD | Acc | WD |
| LoRA | 0.4608 | **0.1877** | 0.4088 | 0.4380 | 0.4189 | 0.4038 | **0.5000** | 0.1667 |
| OPPU | 0.4164 | 0.2287 | **0.4891** | **0.2336** | 0.4189 | 0.2226 | **0.5000** | **0.1667** |
| Cu | 0.4710 | 0.2014 | 0.4526 | 0.3285 | **0.4340** | 0.2792 | **0.5000** | 0.5000 |
| PuQu | 0.4676 | **0.1877** | 0.4599 | 0.3358 | **0.4340** | 0.2830 | **0.5000** | 0.5000 |
| Aplaud+ | **0.4778** | 0.2048 | 0.4672 | 0.2628 | 0.4377 | **0.2038** | **0.5000** | **0.1667** |

## D EXPERIMENTS DETAILS

In this section, we describe the experimental framework used to simulate responses to human surveys using large language models (LLM). Specifically, we present the prompt designs, user profile extraction strategies, and training procedures adopted in our method. Our goal is to enable LLMs to approximate individual-level human responses through structured personalization, achieved via APlaud, a parameter-efficient method that requires orders of magnitude fewer parameters per user.

### D.1 PROMPTS FOR USER PROFILE

To simulate natural-language user profiles for survey response modeling, we construct textual summaries that integrate demographic metadata and selected survey responses from ATP data. These profiles serve as personalized inputs for downstream machine learning tasks, such as response generation or classification. Each summary captures a user's background, financial stressors, and attitudes toward government responsibility. This profile-based approach allows language models to produce outputs that are grounded in realistic user context, improving both personalization and interpretability.

---

**Prompt Template for Simulated User Profile**

**You are a professional assistant tasked with summarizing a user's demographic characteristics and their economic attitudes, financial stressors, and beliefs about inequality and government responsibility based on W54 survey data. Your output should be a single, coherent paragraph suitable for input into a machine learning model.**

**Instructions:**

- Write in complete, natural English sentences.
- Begin by summarizing demographic information: age, sex, race, education, marital status, religion, religious attendance, political party, political ideology, income, and region.
- Then summarize the user's reported financial well-being, including current household finances, experiences growing up, and ability to meet basic needs.
- Include financial worries such as debt, retirement savings, or healthcare expenses.
- Describe the user's access to financial resources and assets, such as savings accounts, investments, or loans.
- Capture beliefs about economic fairness, hard work, and the role of government in providing housing, healthcare, education, or other forms of support.
- Summarize attitudes toward economic inequality—its perceived causes, who is responsible for fixing it, and which policy proposals are seen as effective.
- Include how the user thinks current economic conditions impact various groups (e.g., middle class, wealthy, poor).
- If available, mention expected future economic conditions and views on powerful actors like corporations or wealthy individuals.

> - Skip any questions answered with "Refused" or missing responses.
> - Do not add interpretation or sentiment beyond what is explicitly stated.
>
> **User demographic metadata:** `{{metadata}}`
> **Survey responses:** `{{profile_questions}}` **Generate a concise, fluent paragraph summarizing the user:**

## D.2 PROMPTS FOR USER PROFILE AND HISTORY Q&A

We extend the simulated user profiling approach by merging each generated profile with a subset of previously answered survey questions and responses. This combined context served as an input prompt to simulate responses to new, unseen test questions. The prompt includes three key components: (1) a natural-language user background summary generated from structured metadata and survey answers, (2) the new survey question to be predicted, and (3) a multiple-choice format with clear answer options. The prompt is explicitly designed to constrain the model's output to a single valid choice (e.g., A, B, C), allowing for consistent evaluation and comparison across users and items. This approach allowed the language model to condition its predictions on both the inferred user profile and their past answer behavior, enhancing personalization and response coherence.

> **Prompt Template: Simulated Survey Response with Profile + History**
>
> **You are user {{user_id}}, with the following background summary:**
> {{user_profile_paragraph}}
>
> **Here is the question:**
> {{test_question}}
>
> **This is a single-answer multiple choice question. Here are the options:**
> {{A. ..., B. ..., C. ..., etc.}}
>
> **Please select the most appropriate answer based on your background.**
> Respond with only the corresponding uppercase letter (e.g., A, B, C), and format your answer exactly like this: A
> Do not include any explanation, reasoning, or repeat the question.

## D.3 DATA AND PROMPTS FOR GENERATING USER PROFILE

We generate each user profile using **ChatGPT-4**. The profile is constructed from two sources of information:

- **Survey-provided demographic metadata.** These metadata fields come directly from the original survey and include:
    - Region (CREGION): Northeast, Midwest, South, West
    - Sex (SEX): Male, Female
    - Age group (AGE): 18–29, 30–49, 50–64, 65+
    - Education level (EDUCATION): Less than high school, High school graduate, Some college, Associate's degree, College graduate, Postgraduate
    - Citizenship (CITIZEN): Yes, No
    - Marital status (MARITAL): Married, Divorced, Separated, Widowed, Never married
    - Religion (RELIG): Protestant, Catholic, Jewish, Muslim, Buddhist, Hindu, Atheist, Agnostic, Other, Nothing in particular
    - Religious attendance (RELIGATTEND): More than once a week, Weekly, Monthly, Few times/year, Seldom, Never
    - Political party (POLPARTY): Republican, Democrat, Independent, Other
    - Political ideology (POLIDEOLOGY): Very conservative, Conservative, Moderate, Liberal, Very liberal

– Race/ethnicity (RACE): White, Black, Asian, Hispanic, Other
– Income (INCOME): $< 30k, 30 - 50k, 50 - 75k, 75 - 100k, > 100k$

- **10 survey questions most relevant to user characterization**. These are selected among the user's answered items and reflect personal attitudes, preferences, or values. To avoid information leakage, we **remove** these 10 profile-related questions prior to constructing the train/validation/test split.

The generated profile is a neutral paragraph rewriting the demographic metadata and the selected 10 questions. No additional information is inferred.

The exact prompt used to generate the profile is shown below.

---

**Prompt for Generating the User Profile**

**You are a professional assistant tasked with summarizing a user's demographic information and survey response profile in a clean, coherent paragraph for input into a machine learning model.**

**Instructions:**

- Use complete, natural English sentences.
- Start by summarizing demographic information (age, sex, race, education, marital status, religion, political ideology, income, device type, language).
- Then summarize the user's self-reported life satisfaction.
- Then summarize their leadership values and views about business or political leadership, based only on answered questions.
- Then summarize their beliefs about gender and leadership, if any.
- Skip any survey questions where the user answered "No answer."
- Be neutral and descriptive, without adding interpretation.

**User demographic metadata:**
{metadata}

**Survey responses (10 most profile-relevant items):**
{profile_text}

**Generate a concise, fluent paragraph summarizing the user.**

---

# E    DETAILS ON HUMAN STUDIES DATA: PEW ATP AND GENERAL SOCIETY SURVEY

## E.1    PEW RESEARCH ATP

The American Trends Panel (ATP) is a nationally representative panel of U.S. adults conducted by the Pew Research Center. ATP is designed to study a wide variety of topics, including politics, religion, internet usage, and family life. We analyze sampled questions from four waves, selecting only *ASK ALL* questions—that is, questions posed to all respondents regardless of subgroup membership or branching logic. In the original ATP design, many questions include randomized Likert-scale options (e.g., positive-to-negative or vice versa). To align with this, we also randomize the presentation order of answer choices in our LLM prompts.

### E.1.1    ATP WAVE 36

Wave 36 (fielded June 19 – July 2, 2018) explores public attitudes toward gender representation in leadership roles. While a majority of Americans express support for having more women in top leadership positions, many remain skeptical that gender parity will be achieved. Views vary notably by political affiliation and gender, reflecting broader social divides.

**Sample Questions from ATP Wave 36**

**Q1.** In general, how important, if at all, is it to you for someone in a top executive business position to provide guidance or mentorship to young employees?
*Options:* (A) Essential    (B) Important, but not essential    (C) Not important    (D) Refused

**Q2.** Do you think that men and women in leadership roles are...
*Options:* (A) Basically similar    (B) Basically different    (C) Refused

**Q3.** Who generally has a better approach to leadership?
*Options:* (A) Women    (B) Men    (C) Neither    (D) Refused

**Q4.** What is the ideal situation for the number of women in high political office?
*Options:* (A) More, but still fewer than men    (B) Equal    (C) More than men    (D) Refused

**Q5.** What is the ideal number of women in top executive business positions?
*Options:* (A) More, but still fewer than men    (B) Equal    (C) More than men    (D) Refused

**Q6.** As more women run for office...
*Options:* (A) Gender parity is inevitable    (B) Men will still dominate    (C) Refused

**Q7.** As more women enter management...
*Options:* (A) Gender parity is inevitable    (B) Men will still dominate    (C) Refused

**Q8.** How much would more women in leadership improve life for women?
*Options:* (A) A lot    (B) Some    (C) Not much    (D) Nothing    (E) Refused

**Q9.** How much would more women in leadership improve life for men?
*Options:* (A) A lot    (B) Some    (C) Not much    (D) Nothing    (E) Refused

**Q10.** How much would more women in leadership improve life for all Americans?
*Options:* (A) A lot    (B) Some    (C) Not much    (D) Nothing    (E) Refused

ATP WAVE 42

Wave 42 of the American Trends Panel, conducted from January 7 to January 21, 2019, focuses on public attitudes toward scientists, trust in science, and perceptions of the scientific method. The survey explores how Americans view the role of scientists in public policy, their confidence in scientific experts, and whether science is seen as a force for societal good. Respondents were also asked about the objectivity and integrity of scientists, as well as how much trust they place in scientists from different institutional backgrounds (e.g., industry, government, academia). The data provide insight into partisan and demographic divisions in trust toward scientific information and decision-making processes.

**Sample Questions from ATP Wave 42**

**Q1.** Compared with twenty years ago, do you think developments in science have made people's lives...
*Options:* (A) Better    (B) Worse    (C) About the same

**Q2.** Looking ahead to the next twenty years, do you think developments in science will make people's lives...
*Options:* (A) Better    (B) Worse    (C) About the same

**Q3.** Overall, would you say science has had a mostly positive effect on our society or a mostly negative effect on our society?
*Options:* (A) Mostly positive    (B) Mostly negative    (C) Equal positive and negative effects

**Q4.** How much confidence, if any, do you have in scientists to act in the best interests of the public?
*Options:* (A) A great deal    (B) A fair amount    (C) Not too much    (D) No confidence at all

**Q5.** Which of these statements comes closer to your own view?
*Options:* (A) Scientists should take an active role in public policy debates
(B) Scientists should stay out of public policy debates

**Q6.** Which of these statements comes closer to your own view?
*Options:* (A) Public opinion should guide scientific policy
(B) Issues are too complex for public opinion to guide

**Q7.** In general, would you say scientific experts are...
*Options:* (A) Usually better    (B) Usually worse    (C) Neither

**Q8.** When you hear research is reviewed by an independent committee, does this make you...
*Options:* (A) Trust more    (B) Less    (C) No difference

**Q9.** Which best describes what you think about the scientific method?
*Options:* (A) Accurate conclusions    (B) Can produce any desired conclusion

**Q10.** Which of these statements comes closer to your view?
*Options:* (A) Judgments based solely on facts    (B) Judgments as biased as others'

ATP WAVE 50

Wave 50 of the American Trends Panel was conducted from June 25 to July 8, 2019, with responses from 9,834 U.S. adults. This wave focused on family life, romantic relationships, parenting, cohabitation, marriage expectations, and household dynamics. The survey included split-form designs to compare attitudes toward men and women across different relationship and parenting roles. Questions also explored satisfaction with family life, financial situations, and perceived social support. Responses were collected online, with weighting applied to ensure national representativeness across demographics such as age, gender, race, education, political affiliation, and internet access.

### Sample Questions from ATP Wave 50

**Q1.** In general, how important is it for a **man** to have a job or career he enjoys in order to live a fulfilling life?
*Options:* (A) Essential    (B) Important, but not essential    (C) Not important

**Q2.** In general, how important is it for a **woman** to have a job or career she enjoys in order to live a fulfilling life?
*Options:* (A) Essential    (B) Important, but not essential    (C) Not important

**Q3.** What do you think is the ideal situation for **women with young children**?
*Options:* (A) Working full-time    (B) Working part-time    (C) Not working for pay

**Q4.** What do you think is the ideal situation for **men with young children**?
*Options:* (A) Working full-time    (B) Working part-time    (C) Not working for pay

**Q5.** Do you think **couples who live together before marriage** have a...
*Options:* (A) Better chance at a successful marriage    (B) Worse chance    (C) Doesn't make much difference

**Q6.** How much pressure, if any, do you feel from **society** to marry your partner?
*Options:* (A) A lot    (B) Some    (C) Not too much    (D) No pressure at all

**Q7.** How do you feel about the way **household chores** are divided between you and your partner?
*Options:* (A) Very satisfied    (B) Somewhat satisfied    (C) Somewhat dissatisfied    (D) Very dissatisfied

**Q8.** Have you ever **reduced your work hours** due to balancing parenting and career?
*Options:* (A) Yes    (B) No

**Q9.** Do you think couples who are **not married but living together** can raise children as well as married couples?
*Options:* (A) Yes    (B) No

**Q10.** Do you trust your partner to **handle money responsibly**?
*Options:* (A) A great deal    (B) A fair amount    (C) Not much    (D) Not at all

ATP WAVE 54

Wave 54 of the American Trends Panel was conducted from September 16 to 29, 2019, with responses from 6,878 U.S. adults. This wave focused on attitudes toward gender roles, parenting, household responsibilities, and societal expectations. Respondents were sampled across five strata to improve representation of underrepresented groups. The survey was administered online, with weights applied to correct for demographic and behavioral differences. The margin of error for the weighted sample is ±1.59 percentage points.

### Sample Questions from ATP Wave 54

**Q1.** Would you say there is...
*Options:* (A) Too much economic inequality    (B) Too little economic inequality    (C) About the right amount

**Q2.** Do you think the U.S. economic system...
*Options:* (A) Requires only minor changes    (B) Requires major changes    (C) Needs to be completely rebuilt

**Q3.** How much responsibility should the federal government have in reducing economic inequality?
*Options:* (A) A lot    (B) Some    (C) Only a little    (D) None

**Q4.** How much does the current tax system contribute to economic inequality?
*Options:* (A) A great deal    (B) A fair amount    (C) Not too much    (D) Not at all

**Q5.** Do you think some people start out with more opportunities than others?
*Options:* (A) Contributes a great deal to inequality    (B) A fair amount    (C) Not too much    (D) Not at all

**Q6.** How much would raising the federal minimum wage reduce economic inequality?
*Options:* (A) A great deal    (B) A fair amount    (C) Not too much    (D) Nothing at all

**Q7.** How much would expanding Medicare to cover all Americans reduce economic inequality?
*Options:* (A) A great deal    (B) A fair amount    (C) Not too much    (D) Nothing at all

**Q8.** Should the government invest in education and job training, or give direct financial assistance?
*Options:* (A) Invest in education and job training    (B) Give direct assistance

**Q9.** Do you think filling out the U.S. census will...
*Options:* (A) Benefit you personally    (B) Harm you personally    (C) Neither benefit nor harm

**Q10.** How important is it for the government to provide a high-quality K–12 education?
*Options:* (A) Yes, it's the government's responsibility    (B) No, it's not

**Q11.** Thinking about your household's financial situation, how much are you affected by job availability in your area?
*Options:* (A) A great deal    (B) A fair amount    (C) Not too much    (D) Not at all

**Q12.** How often do you worry about the cost of health care?
*Options:* (A) Every day    (B) Almost every day    (C) Sometimes    (D) Rarely    (E) Never

**Q13.** Have you received government assistance such as SNAP, Medicaid, or unemployment benefits in the past 12 months?
*Options:* (A) Yes    (B) No

**Q14.** How much does your family's financial situation affect your children's ability to succeed in life?
*Options:* (A) A great deal    (B) A fair amount    (C) Not too much    (D) Not at all

**Q15.** How do you rate current U.S. economic conditions?
*Options:* (A) Excellent    (B) Good    (C) Only fair    (D) Poor

GENERAL SOCIAL SURVEY (GSS)

The General Social Survey (GSS) 2016–2020 Panel is a longitudinal dataset that re-interviewed respondents from the 2016 and 2018 GSS cross-sectional samples to measure social and attitudinal change over time. Participants from these earlier waves were invited to complete a follow-up survey in 2020. The resulting three-wave panel study includes responses from 2016 (Wave 1a), 2018 (Wave 1b), and 2020 (Wave 2).

---

**Sample Questions from the GSS Panel**

**Q1.** What do you think the chances are these days that a white person won't get a job or promotion while an equally or less qualified Black person gets one instead?
*Options:* 1 = Very likely; 2 = Somewhat likely; 3 = Not very likely

**Q2.** In general, do you think the courts in this area deal too harshly or not harshly enough with criminals?
*Options:* 1 = Too harshly; 2 = Not harshly enough; 3 = About right

**Q3.** Should divorce in this country be easier or more difficult to obtain than it is now?
*Options:* 1 = Easier; 2 = More difficult; 3 = Stay as is

**Q4.** Do you feel that the demands of your family life interfere with your job?
*Options:* 1 = Always; 2 = Often; 3 = Sometimes; 4 = Hardly ever; 5 = Never

**Q5.** Have you ever given up or would you give up good job opportunities for the benefit of your family life?
*Options:* 1 = Yes, and would again; 2 = Yes, but wouldn't again; 3 = No, but would; 4 = No, and wouldn't

**Q6.** A working mother can establish just as warm and secure a relationship with her children as a mother who does not work.
*Options:* 1 = Strongly agree; 2 = Agree; 3 = Disagree; 4 = Strongly disagree

**Q7.** It is much better for everyone involved if the man is the achiever outside the home and the woman takes care of the home and family.
*Options:* 1 = Strongly agree; 2 = Agree; 3 = Disagree; 4 = Strongly disagree

**Q8.** Because of past discrimination, employers should make special efforts to hire and promote qualified women.
*Options:* 1 = Strongly agree; 2 = Agree; 3 = Neither; 4 = Disagree; 5 = Strongly disagree

**Q9.** Do you favor or oppose preferential hiring and promotion of women?
*Options:* 1 = Strongly favor; 2 = Not strongly favor; 3 = Not strongly oppose; 4 = Strongly oppose

**Q10.** Most men are better suited emotionally for politics than are most women.
*Options:* 1 = Agree; 2 = Disagree

**Q11.** A preschool child is likely to suffer if his or her mother works.
*Options:* 1 = Strongly agree; 2 = Agree; 3 = Disagree; 4 = Strongly disagree

**Q12.** Should the government promote equality between men and women?
*Options:* 1 = Definitely should; 2 = Probably should; 3 = Probably should not; 4 = Definitely should not

**Q13.** Compared with American families in general, how would you rate your family income?
*Options:* 1 = Far below average; 2 = Below average; 3 = Average; 4 = Above average; 5 = Far above average

**Q14.** Are you satisfied with your present financial situation?
*Options:* 1 = Pretty well satisfied; 2 = More or less satisfied; 3 = Not satisfied at all

**Q15.** Do you think there is any area near here where you would be afraid to walk alone at night?
*Options:* 1 = Yes; 2 = No

## F  RELATION WITH RECOMMENDER SYSTEM

Personalization has long been a central theme in recommender systems, where models infer user preferences from historical interactions and estimate item relevance over a large catalog. Conceptually, survey QA prediction has a distant parallel to collaborative filtering. When responses are binarized (yes/no or agree/disagree), a survey can be represented as a $Respondents \times Items$ matrix where each entry reflects a respondent's position. For multi-level items such as Likert scales, each question–response option can be expanded into a set of binary indicators–for example, mapping a 5-point item into five item-specific binary variables – yielding a uniform binary representation across all items. Alternatively, when responses reflect ordered categories, these items may be encoded using a single ordinal score (e.g., 1–5), which preserves the inherent ordering of the response levels. Together, these encoding strategies allow heterogeneous survey instruments to be transformed into a structured matrix format that is compatible with downstream modeling. From this perspective, predicting a respondent's answer to a new item resembles preference completion in recommender systems.

Despite this superficial similarity, the underlying formulation is fundamentally different. Modern recommendation is typically framed as a learning-to-rank problem, whereas personalized survey response prediction does not involve ranking. A second key distinction concerns the observation regime: recommender systems operate under extreme sparsity, where each user interacts with only a tiny fraction of the item space, and models must infer preferences from partial interactions across many users. In contrast, survey datasets provide complete responses over a shared set of questions, and user preference is inferred directly from how individuals semantically interpret and answer natural-language survey items. As a result, our problem is closer to modeling user-specific semantic judgments than to reconstructing latent preference structures from sparse interactions.

Classical recommendation methods such as matrix factorization(32), NeuMF(31), or LightGCN(30) rely exclusively on latent collaborative filtering signals without any item semantics. These approaches treat items as no-smenantic indices and assume user–item interactions follow a modeled structure. Such assumptions do not hold in our setting: survey questions have explicit wording and domain meaning, and since every user observes the same question set, collaborative filtering cannot exploit sparsity or cross-user item co-occurrence patterns, nor can it model the semantic structure shared across questions. As a result, traditional recommendation techniques are not directly applicable to personalized survey response generation.

More recently, large language models have been incorporated into recommender systems. Early studies (59; 13; 84) explore LLM's zero-shot/few-shot potential via in-context learning. The mismatch between LLMs' general-purpose training and the specific demands of recommendation tasks results in inadequate performance. To better align LLM with the recommendation domain, one research line would formulate recommendation as a sequential generation task and methods such as P5 (23), M6-Rec (12) and serialize user–item histories into natural-language prompts and train an LLM to generate the next item or a ranked list. Another research line would use LLMs as auxiliary modules to enrich representations which leverages LLMs to augment item/user embeddings or to support re-ranking such as GPT4Rec (52), LaMAR (77), LLM4Rec (21). To the best of our knowledge, existing LLM-based recommenders generally adopt a one-size-fits-all design and compress the personalized information into input tokens either by hard or soft prompts. Thus, while conceptually related through the lens of personalization, our framework achieves personalization model-wise and introduces a parameter-efficient personalization not present in current recommender system literature.

Overall, our contribution is orthogonal to the development of recommendation systems. We introduce (i) an SVD-based *shared subspace* $(U, V)$ tailored specifically for structured survey QA, and (ii) a *per-user lightweight personalization layer* $(C_u, \alpha_u, \beta_u)$ learned within that subspace. This design directly captures individual answer preferences under a shared question structure. To the best of our knowledge, no existing recommendation method performs LLM-powered low-rank shared-subspace personalization learning, making our approach distinct in both problem setting and technical design.

## G  LIMITATION

Our evaluation is confined to a limited set of survey datasets, necessitating future work to validate generalization across more domains and populations. APlaud's per-user personalization relies on the quantity and quality of historical responses. Users with very limited past data (e.g., fewer than 20-40 questions) may experience less robust personalization, potentially leading to overfitting or reduced accuracy. While effective for survey prediction, APlaud's direct applicability to other personalized LLM tasks (e.g., personalized text generation, conversational agents) without further adaptation remains relatively unexplored.

## H  LLM USAGE STATEMENT

We used ChatGPT solely for polishing writing at the sentence and paragraph level. The content and contributions of this paper were created by the authors. All text refined with ChatGPT has been carefully checked to avoid errors.

