# OpenReview forum: "APlaud: Adaptive Personalized Low-Rank Decomposition for User-Specific LLM"
_ICLR.cc/2026/Conference — Submitted to ICLR 2026_

### Official Review · Reviewer_broL · 2025-10-15

**Soundness:** 3
**Presentation:** 4
**Contribution:** 3
**Rating:** 8
**Confidence:** 3

**Summary:**

The paper proposes APlaud, a lightweight framework for personalizing LLMs to individual users without maintaining a full adapter per user. It first learns a shared adapter from all users, then decomposes it into common patterns and small user-specific components. Each user only stores a tiny correction layer that adjusts the shared representation, making personalization memory-efficient. A variant APlaud+ compresses these user modules even further.

**Strengths:**

- APlaud introduces a well-structured approach that distinguishes between global model knowledge shared across users and fine-grained user-specific differences. By reusing a shared representation and learning only small adjustments per user, the method effectively captures personalization while staying lightweight.
- The framework achieves major savings in storage and computational cost compared to maintaining a separate adapter for each user. The results demonstrate that even when thousands of users are supported, the added storage and serving cost remain minimal.

**Weaknesses:**

- Assumption that the shared subspace is “stable” may be fragile. In particular, claiming that U,V are “relatively stable across users” are plausible for shared question structure, but the paper doesn’t measure subspace drift across waves, topics, or time (ATP waves differ materially). If drift is non-trivial, freezing U,V could bake in population bias.
- Inconsistent notations: Ablation Table 4 labels “TS” as Twitter Stance in the caption while the main text defines TS = Trust in Science, but this is rather a minor problem.
- While storage and parameter counts are favorable, serving at million-user scale depends on how quickly per-user heads can be fetched/instantiated per request across multiple layers. The paper can maybe discuss a little about the latency/throughput under realistic multiplexing (e.g., cache hit rates, cold-start users).

**Questions:**

- What exactly is included in the LLM-generated user profile (the 30% question subset)? Could you discuss the results where no profile is used, or where profiles are built from non-overlapping meta-questions only?
- How stable are the learned U,V across waves/topics/time?

---

> ### Author Response · Authors · 2025-11-23
>
> We sincerely thank reviewer broL for the positive and constructive evaluation. We appreciate the recognition of APlaud as a clear, lightweight, and scalable personalization framework. In the revision and responses below, we added a CKA-based analysis showing the stability of the shared subspace across topics, discussed serving scalability, and clarified user profile details. We are grateful for the reviewer’s feedback, which helped us strengthen both the empirical validation and practical relevance of our work.
>
> > W1& Q2: Stablization of shared U, V
>
> We appreciate the reviewer’s concern about subspace stability. To directly address this, we added empirical validation measuring subspace drift across topics in the revised manuscript.
>
> We computed *Centered Kernel Alignment* (CKA)[1] similarities of the SVD components $U$ and $V$ between two distinct topics (G&L vs. TS), shown in Fig.4 of the revised manuscript. The results reveal complementary stability patterns: $V$ maintains high cross-topic consistency (CKA~$\approx$~0.96), functioning as a stable coordinate generator, while $U$ shows moderate variation (CKA~$\approx$~0.20) that captures topic-specific semantics. This demonstrates that our shared low-rank decomposition remains well-aligned across topics, with sufficient structural regularity to support the decomposition.
>
> Specifically, we note that survey data possess unique characteristics that naturally align with our architectural assumptions. Survey organizations intentionally design questionnaires with semantically consistent questions across cohorts and waves to enable comparable measurements over time—critical since re-contacting respondents is costly. They carefully control sampling frames to maintain coherent populations and account for temporal drift in rapidly changing domains. Within any given survey, questions presented to users are typically identical or highly overlapping, while users differ in their responses. This structural regularity directly motivates our separation of \textit{survey content} (modeled via shared $U,V$) from *user preference* (modeled via personalized $C_u, \alpha_u, \beta_u$).
>
> Importantly, our method does not assume a universal $(U,V)$ pair generalizes across all topics or time periods. The model learns separate $(U,V)$ bases for each topic, ensuring personalization operates within an appropriately aligned subspace specific to that survey domain. This design acknowledges that different topics may have distinct semantic structures while leveraging the within-topic regularity characteristic of well-designed survey instruments.
>
> In practice, survey organizations are fully aware of temporal and cohort limitations—they do not anticipate asking follow-up questions to different respondents across waves, nor do they expect a single model to generalize indefinitely. Our approach mirrors standard practice in recommender systems, where models are periodically re-estimated when meaningful drift or population shifts occur. Similarly, when applying our method across new survey waves or substantially different cohorts, practitioners would naturally re-train $U,V$ to reflect new semantics or population composition. This is not a limitation of our approach but an accurate reflection of how survey modeling is operationalized in practice.
>
> > W2: Inconsistent notation
>
> We thank the reviewer for pointing out the inconsistency in the notation of TS. We have corrected this typo in the revised revision.
>
>
>
>
>
> Refrences:
> 1. Kornblith, Simon, et al. "Similarity of neural network representations revisited." International conference on machine learning. PMlR, 2019.

---

> ### Author Response · Authors · 2025-11-23
>
> >W3: Serving hinges on per-head fetch latency, throughput under realistic multiplexing
>
> We appreciate the reviewer’s attention to practical deployment concerns at scale. Our architecture’s compact parametric design directly facilitates efficient serving.
>
> Each user is represented by compact parameters $(C_u, \alpha_u, \beta_u)$, typically less than 1~MB in total across all layers. The shared subspace $(U,V)$ remains fixed in GPU memory, and only these lightweight user-specific components need to be fetched per request.
>
> Also, several aspects of our design and typical survey workloads support scalable and efficient serving:
> - *High cache efficiency*: Survey participation follows predictable campaign windows, so only a small subset of users are active at any given time. Caching parameters for the top 10K active users ($\approx$10~GB) can cover most requests in large-scale deployments.
> - *Question-level caching synergy*: Since all users answer identical questions within a wave, shared subspace computations ($U,V$) and question-side representations can be cached and reused across users, significantly reducing redundant computation.
> - *Prefetching opportunities*: Surveys typically provide advance notice of expected participants; user parameters can be prefetched into cache before responses arrive, eliminating cold-start latency in most cases.
> - *Graceful cold-start handling*: For cache misses, the base shared model ($U,V$) can serve with minimal degradation, while personalized parameters are asynchronously loaded for subsequent interactions.
> - *Production compatibility*: Modern LLM serving frameworks such as vLLM already support efficient KV-cache management and user-level parameter isolation, and our compact user modules integrate naturally with such architectures. The per-user parameter size is comparable to (or smaller than) user embeddings commonly used in large-scale recommendation systems.
>
> > Q1: Details of LLM-generated user profile and discuss on results where no profile is used
>
> We thank the reviewer for this insightful question regarding user profile construction and whether our method depends critically on profile information.
>
> **Profile construction details**: In the revised manuscript, we include the full prompt template used to generate each user profile (Appendix D.3). Each profile summarizes two sources of information: (1) survey-provided demographic metadata (e.g., age, gender, education), and (2) responses to the ten most profile-relevant survey questions, rewritten into a neutral descriptive paragraph. Importantly, these profile questions are removed from the questions used for training or evaluation, ensuring that profile information does not leak target answers.
>
> To test whether personalization depends critically on profiles, we conducted an ablation experiment removing the entire profile paragraph from prompts while keeping all other settings identical. Results in T1 (Table 7) of the revised manuscript show that removing profile information causes a consistent performance drop across all methods, confirming that user profiles provide valuable preference cues.
>
> T2: Performance comparison with profile being removed from prompts with llama2-7B backbone
>
> | Method  | G&L |  | TS |  | F&R |  | EI |  | GSS |  |
> |--------|:---:|:---:|:---:|:---:|:---:|:---:|:---:|:---:|:---:|:---:|
> |        | ACC | F1 | ACC | F1 | ACC | F1 | ACC | F1 | ACC | F1 |
> | LoRA   | 0.4576 | 0.3106 | 0.6454 | 0.3249 | 0.5486 | 0.2598 | 0.4725 | 0.3028 | 0.3583 | 0.2139 |
> | OPPU   | 0.6271 | 0.6030 | 0.7528 | 0.6115 | 0.5970 | 0.3462 | 0.4942 | **0.4005** | 0.3686 | 0.2322 |
> | Cu     | 0.6151 | 0.5798 | 0.6991 | 0.5838 | 0.5681 | 0.2975 | 0.4942 | 0.3602 | 0.3743 | 0.2545 |
> | Aplaud | **0.6457** | **0.6225** | 0.7599 | 0.6165 | 0.6148 | 0.3498 | 0.4942 | 0.3614 | 0.3743 | 0.2604 |
> | Aplaud+| 0.6329 | 0.6079 | **0.7639** | **0.6231** | **0.6200** | **0.3599** | **0.5083** | 0.3918 | **0.3757** | **0.2617** |
>
> Crucially, however, **APlaud** and **APlaud+** remain highly competitive and often surpass **OPPU** across multiple datasets even without profiles—despite using significantly fewer per-user parameters (e.g., 0.24M vs. 134M for Llama-3-8B). This demonstrates that our lightweight parametric modules effectively capture stable user preference patterns directly from response histories without requiring explicit profile features.
>
> These results highlight two major advantages of our approach:
> (1) *Robustness*: performance degrades gracefully when profile information is unavailable, rather than collapsing entirely; and  (2) *Parameter efficiency*: our method achieves strong personalization with orders of magnitude fewer parameters than baseline approaches, whether or not profiles are used.
> In deployment scenarios where demographic metadata may be incomplete, sensitive, or unavailable, **APlaud** can still deliver effective personalization through compact, learned parametric representations alone.

---

> ### Comment · Reviewer_broL · 2025-11-27
>
> Thanks to the authors for addressing my concerns. I believe my score is high enough and good luck with the submission.

---

### Official Review · Reviewer_oHND · 2025-10-30

**Soundness:** 2
**Presentation:** 3
**Contribution:** 2
**Rating:** 2
**Confidence:** 2

**Summary:**

This paper introduces a technique called APlaud (Adaptive Personalized Low-rank and User-specific Nested Decomposition), for personalized survey response prediction using fine-tuned LLMs. The paper discusses about synthetic response generation for surveys which appear like real human responses. There are some existing works in this domain but have few issues: a) works which involve LLM prompt strategies show some cultural biases b) works which involve fine-tuning (LoRA based) are able to generate model responses at sub-population level and cannot model user response at individual level. The paper claims that the amount of data available per user is often much smaller than the size of the personalised parameters (overfitting possible), the number of users could be very large, so training separate LoRA for each not very feasible and the users should not be treated in isolation, as survey involves same question across users there is a semantic correlation between the users. APlaud extends the LoRA paradigm by combining a shared low-rank basis with compact user-specific corrections and residual terms, further reducing parameter costs through nested low-rank factorization. The paper claims that this work is the first to explicitly formulate and study the survey prediction problem in the personalized LLM setting.

**Strengths:**

S1: The paper is well written and organized.

S2: The paper motivates well about the need for synthetic response generation for surveys to aliviet cost - albeit there are some practical drawbacks or lack of clarity in the solution approch.

S3: The paper evalated the technique on a broad set of datasets and presents detailed ablation studies including noise injection.

**Weaknesses:**

W1:  It is unclear that in real deployment situation - who will be creating these digital-twins. It is not practical that users themselves would be interested in doing these. So that practicality of the solution remains questionable.

W2: Although the papers discusses about learning low rank matrices for all users, this could still be expensive for very large number of users.

W3: Since this work is about personalisation, a comprehensive user survey could be useful.

W4: The paper does not provide details on missing or incomplete responses.

W5: The approach has similarities with recommender systems - some dedicated discussion is need to compare the novelty of the proposed technique compared to recommendation systems literature.

**Questions:**

Q1: Method seems very restrictive to synthetic survey response generation. Whether the research problems need to changed drastically to use it for other open-ended tasks ?

Q2: Most baseline are adapter/LoRA-style family based. Comparing the method against non-adapter based could be good such as methods like retrieval based (memory maintained for users and relevant context retrieved for response generation) ?

---

> ### Author Response · Authors · 2025-11-23
>
> We thank the reviewer for carefully reading our work and for recognizing its organization, motivation, and comprehensive experimental analysis. While we appreciate these positive remarks, we respectfully note that several concerns seem to arise from a misunderstanding of the paper’s goals and scope. Our work does not attempt to build "digital twins" in a deployment context, nor to replace human survey participants. Rather, the paper focuses on the *problem formulation and model innovation*, introducing **APlaud**, the first principled framework for *personalized survey response prediction* under strict parameter- and data-efficiency constraints.
>
> In contrast to prior LLM personalization or recommender approaches, APlaud explicitly exploits the structural regularity of survey data—where questions are consistent across users—to decouple shared semantic factors from individual preference factors. This allows large-scale personalization with a fraction of the parameters required by existing LoRA-based or memory-based methods. Importantly, our formulation is general and can extend beyond survey QA to open-ended or generative tasks, as demonstrated by our additional experiment on the LAMP News Headline generation benchmark.
>
> We also emphasize that APlaud is not intended as a commercial deployment framework but as a *methodological foundation* for efficient, scalable, and interpretable personalization. We have clarified these distinctions in the revision, added discussion contrasting recommender-style factorization, and provided new analyses (Appendix F and Table 9) confirming that our two-stage design cleanly separates shared and personalized learning. We believe these clarifications and new results address the reviewer’s concerns and strengthen the contribution of the paper.
>
> > W1: Real deployment situation and practicality of solution.
>
> We respectfully disagree that the practicality of our approach is questionable. Importantly, the entities creating these models are not individual end-users but the organizations that conduct surveys and own the underlying data. Moreover, our work does not aim to build full-scale “digital twins” of individuals; rather, it focuses on developing lightweight, data-driven *personalized synthetic respondents* that can represent user-level response patterns within controlled survey contexts.
>
> - **Who creates them**: The natural creators and users of such personalized synthetic respondents include:
> (1) *Professional survey organizations* (e.g., Pew Research, GSS, Gallup, ANES);
> (2) *Marketing and market research agencies* that maintain proprietary respondent panels; and
> (3) *Enterprises* conducting customer satisfaction, employee engagement, or market segmentation studies.
> These entities already possess the necessary data, infrastructure, and strong economic incentives to deploy such personalized synthetic respondents. In addition, third-party AI service vendors can facilitate co-development or self-service platforms, enabling organizations to adopt our framework securely without requiring extensive in-house AI infrastructure.
>
> - **Real-world motivation**: Our research was motivated by direct industry collaborations that identified a critical operational challenge: posing follow-up questions to respondents who are unavailable or prohibitively expensive to re-contact due to attrition, survey fatigue, or rising incentive costs. Personalized synthetic respondents directly address this need by allowing organizations to pose new questions to modeled representations of their original respondents, enabling continuity of analysis without costly re-recruitment.
>
> The global survey and market research industry is a multi-billion-dollar ecosystem already seeking scalable solutions to respondent attrition and cost escalation. Our framework enables organizations to:  (i) answer follow-up questions without costly re-contact;  (ii) reduce recruitment and incentive expenses;  (iii) rapidly test new survey designs; and  (iv) extend panel life through persistent, personalized synthetic respondents.
>
> The practicality of our method is therefore firmly grounded in the real operational needs and workflows of professional survey and market research organizations.

---

> ### Author Response · Authors · 2025-11-23
>
> >W2: Still be expensive for a very large number of users.
>
> We thank the reviewer for raising this concern. Our method is specifically designed to minimize the per-user parameter footprint while maintaining high personalization fidelity. Concretely, \textbf{APlaud} uses only about 1% of the per-user parameters compared to the strong personalization baseline OPPU (Table 3 in manuscript), while achieving comparable or better predictive performance (Table 1 & 2 in manuscript). This represents a substantial reduction in memory requirements and demonstrates clear scalability advantages for large user populations.
>
> To provide practical context, with a rank configuration of $r=8$, each user requires only a few kilobytes of parameters—orders of magnitude smaller than per-user adapters used in existing LoRA-based methods. This allows millions of personalized models to be supported within typical GPU memory constraints, enabling practical deployment at scale.
>
> Moreover, our low-rank nested formulation provides a principled foundation for additional compression strategies, such as hierarchical shared subspaces or adaptive per-user rank allocation, which can further reduce the footprint if needed. However, our current $\sim$100× parameter reduction already delivers scalable personalization in realistic large-population settings, effectively addressing the reviewer’s concern regarding computational and storage cost.
>
> > W3: A comprehensive user survey could be useful.
>
> We thank the reviewer for this suggestion and would like to clarify an important distinction. The "personalization" in this work refers to modeling *individual survey respondents*, not to personalizing an end-user interface or interactive experience. The study was directly motivated by collaborations with organizations addressing a specific operational challenge—generating accurate follow-up responses from respondents who are no longer available.
>
> Our evaluation already uses authentic response data from thousands of real participants (e.g., Pew Research, GSS, Gallup), providing rigorous ground truth for assessing whether synthetic respondents faithfully capture individual-level response patterns. In this context, personalization is evaluated through predictive accuracy and generalization across real survey datasets, as reported in the paper.
>
> A traditional user survey would not be appropriate here, as this is not a consumer-facing system where subjective user satisfaction is the primary measure. Instead, the relevant question is whether the model accurately replicates user-level response behavior—precisely what our empirical results demonstrate.
>
> Future work could indeed explore organizational-level studies to evaluate whether professional survey researchers find personalized synthetic respondents useful within their operational workflows. While such deployment-oriented evaluations fall outside the methodological scope of this paper, the fact that our work emerged from direct industry collaboration already provides practical validation of its real-world relevance and utility.
>
> > W4: Details on missing or incomplete responses.
>
> We thank the reviewer for this question.  Across the four ATP waves used in our study, the total number of survey items varies by design (W36: 139; W42: 129; W50: 127; W54: 115), and respondents are administered different subsets of items, leading to raw item-level missingness between 28\% and 62\%. We retain respondents with at least 10 valid answers (removing only “Refused”), which reduces missingness in every wave (e.g., W36: 62.1\% → 56.1\%; W50: 54.8\% → 34.4\%) and ensures sufficient signal for stable personalization. This preprocessing is consistent with standard practice in recommendation datasets, where extremely sparse users are filtered because they do not provide enough information for reliable modeling. We have added these details in Appendix A.2 the revised manuscript.

---

> ### Author Response · Authors · 2025-11-23
>
> > W5: Relation to Recommender System
>
> We thank the reviewer for this important observation. We have added a detailed comparison in Appendix F (*Relation with Recommender Systems*) of the revised submission. Below, we highlight the key points:
>
> **Core distinction**: While both domains involve personalization, the fundamental tasks differ:
> - **Recommender systems**: Predict which items a user will prefer from a large, sparse catalog (a selection or ranking problem).
> - **Our task**: Predict how each user will answer the same fixed set of survey questions (a response generation problem).
>
> **Key technical differences**:
> - **No sparsity or ranking**: All users see all questions; personalization occurs at the response-content level, not through item selection.
> - **Semantic grounding required**: Survey responses depend on linguistic interpretation and attitude modeling, not merely on user–item interaction patterns.
> - **Structured subspace design**: Our SVD-based shared subspace $(U, V)$ with per-user parameters $(C_u, \alpha_u, \beta_u)$ is specifically designed for structured survey QA under a common question set.
>
> **Relation to existing work**: Traditional RecSys methods (e.g., matrix factorization, NeuMF, LightGCN) treat items as indices without semantic content. Recent LLM-based recommender systems (e.g., P5, GPT4Rec) focus on sequential item generation or textual profile enrichment but do not introduce explicit shared low-rank subspaces or structured per-user adaptation within LLMs for fixed question sets.
>
> **Our novelty**: **APlaud** introduces (i) a shared low-rank subspace tailored for survey response prediction and (ii) an efficient per-user adaptation mechanism—requiring only about 1\% of the parameters used by OPPU—that captures personalized response tendencies under a shared question structure. This formulation is orthogonal to both collaborative-filtering approaches and existing LLM-based recommendation frameworks.
>
> For a detailed technical comparison and discussion , please refer to Appendix F of the revised paper.
>
> > Q1: Extend to other tasks
> We thank the reviewer for this question. While our primary motivation is survey response modeling, **APlaud**'s low-rank personalization mechanism is not restrictive and generalizes naturally to other personalized tasks.
>
> Survey responses allow us to explicitly separate (1) question content and semantics—captured in the shared subspace—from (2) user-specific response tendencies—captured in per-user parameters. This separation is enabled by the structured nature of surveys, where all users answer the same questions. Importantly, this principle extends to other domains where users respond to similar content or prompts.
>
> **New experiments and empirical validation**: In addition to survey QA tasks, we evaluated APlaud on two additional personalized settings:
> - *Personalized classification*: LaMP Movie-Tagging (genre prediction)—results reported in Tables 1–2 in manuscript.
> - *Personalized text generation*: LaMP News-Headline generation—new results reported in T1 (Table 8 of the revised manuscript).
>
> T1: Generation Task Performance on LaMP News Headline Dataset with llama2-7B backbone
> | Method        | R-1    | R-L    |
> |--------------|--------|--------|
> | GPT5-profile | 0.1312 | 0.1177 |
> | OPPU         | 0.1987 | 0.1832 |
> | Cu           | 0.1987 | 0.1825 |
> | Aplaud       | 0.2003 | 0.1838 |
> | Aplaud+      | 0.1987 | 0.1830 |
>
> The new experiments on the *LaMP News Headline Generation* task show that both **APlaud** and **APlaud+** achieve performance comparable to the strong OPPU baseline while using only about **1%** of its per-user parameters.
>
> **Why it generalizes**: The same mechanism that disentangles shared semantics from personalized responses in survey QA also applies to other domains where users engage with similar content:
> - In *movie tagging*, the shared subspace captures genre semantics, while user parameters model individual tagging preferences.
> - In *headline generation*, the shared subspace captures article semantics, while user parameters model personalized writing or selection tendencies.
>
> These results demonstrate that APlaud’s structured low-rank personalization can potentially generalize into a unified, parameter-efficient framework for personalized modeling across both classification and generation tasks, despite its main focus on synthetic survey data.

---

> ### Author Response · Authors · 2025-11-23
>
> > Q2: Comparison with non-adapter methods such as retrieval-based approaches
>
> We thank the reviewer for this helpful suggestion regarding comparisons with non-adapter, retrieval-based personalization methods. In the revised manuscript, we have added two strong GPT-5 baselines to address this point, see T1 (Tables1–2 in the revision submission).
>
> T1: More Comprehensive Performance comparison with zero-shot and retrieval-based baseline with llama2-7B backbone
> | Method | G&L |  | TS |  | F&R |  | EI |  | GSS |  | LAMP MV |  |
> |-------|:---:|:---:|:---:|:---:|:---:|:---:|:---:|:---:|:---:|:---:|:---:|:---:|
> |       | ACC | Macro-F1 | ACC | Macro-F1 | ACC | Macro-F1 | ACC | Macro-F1 | ACC | Macro-F1 | ACC | Macro-F1 |
> | **Non-Personalized** |||||||||||||
> | LoRA      | 0.6393 | 0.6140 | 0.7052 | 0.5343 | 0.5772 | 0.3348 | 0.4617 | 0.3308 | 0.3785 | 0.2479 | 0.6214 | 0.5076 |
> | PiSSA     | 0.6296 | 0.6142 | 0.7386 | 0.5620 | 0.5473 | 0.3427 | 0.4783 | 0.3722 | 0.3559 | 0.2333 | 0.6201 | 0.5280 |
> | MiLoRA    | 0.6714 | 0.6634 | 0.7406 | 0.5768 | 0.5551 | 0.3503 | 0.4618 | 0.3522 | 0.3836 | 0.2681 | 0.6308 | 0.5341 |
> | AdaLoRA   | 0.6700 | **0.6675** | 0.7779 | 0.5965 | 0.5564 | 0.3602 | 0.4716 | 0.3689 | 0.3907 | 0.2834 | 0.6146 | 0.5058 |
> | QLoRA     | 0.6618 | 0.6475 | 0.7467 | 0.5316 | 0.5408 | 0.3411 | 0.4683 | 0.3296 | 0.3738 | 0.2531 | 0.6302 | 0.5253 |
> | **Personalized** |||||||||||||
> | GPT5-profile | 0.5394 | 0.5340 | 0.6170 | 0.3448 | 0.4954 | 0.2879 | 0.4566 | 0.3491 | 0.4689 | 0.2570 | 0.5478 | 0.4507 |
> | GPT5-RAG     | 0.6377 | 0.6306 | 0.7001 | 0.6169 | 0.6174 | 0.3542 | 0.5213 | 0.4117 | **0.5106** | **0.3520** | — | — |
> | OPPU         | 0.6651 | 0.6559 | 0.7548 | 0.6275 | 0.6096 | 0.3612 | 0.5008 | 0.4091 | 0.3701 | 0.2588 | 0.6336 | 0.5147 |
> | Cu           | 0.6651 | 0.6453 | 0.7497 | 0.6423 | 0.5994 | 0.3475 | 0.4975 | 0.3611 | 0.3870 | 0.2590 | 0.6414 | 0.5358 |
> | Aplaud       | 0.6731 | 0.6581 | 0.7761 | 0.6637 | 0.6151 | 0.3669 | 0.5042 | 0.3945 | 0.3912 | 0.2715 | 0.6442 | 0.5366 |
> | Aplaud+      | **0.6828** | 0.6642 | **0.7974** | **0.6824** | **0.6381** | **0.3704** | **0.5183** | **0.4180** | 0.3969 | 0.2719 | **0.6593** | **0.5529** |
>
> First, we include a **GPT-5 (profile-only)** zero-shot setting, where the model is provided only with each user’s demographic and metadata profile.
> Second, we add a **retrieval-augmented** baseline, in which GPT-5 retrieves the five most relevant historical QA pairs and incorporates them into the prompt as few-shot exemplars.
>
> As shown in T1 and the revised results, **APlaud** continues to outperform both the zero-shot profile-only and the retrieval-augmented settings. These results demonstrate that memory-style retrieval alone is insufficient for robust personalization, while APlaud’s structured low-rank personalized adaptation provides substantially stronger and more consistent improvements across datasets.

---

> ### Comment · Reviewer_oHND · 2025-11-25
> **Thanks for your response.**
>
> Thanks for your detailed response.
>
> Thanks for explaining the deployment scenario - that “the entities creating these models are not individual end-users, but the organizations that conduct surveys and own the underlying data.” This was one of my biggest concerns.
>
> I have updated my score - assuming that the authors would include their responses to the following questions in the final version:
>
> Better clarifications regarding the deployment scenario - ideally in the introduction
>
> Details on missing or incomplete responses
>
> Comparison with non-adapter methods such as retrieval-based approaches
>
> Some discussion regarding similarity/dissimilarity with recommender systems

---

> > ### Author Response · Authors · 2025-11-27
> >
> > We sincerely thank the reviewer for the constructive feedback and for raising the score from rejection to weak rejection. We greatly appreciate the reviewer's engagement and the clear roadmap provided for strengthening the paper. Following these suggestions, we have undertaken substantial revisions that directly address all four concerns and materially improve the paper's clarity, rigor, and practical contribution.
> >
> > - **Deployment scenario (Introduction)**.
> > We significantly expanded the introduction (lines 33–37, 70–78, and 131–137) to clearly articulate the real-world deployment setting and practical motivation. The revised text now explicitly distinguishes the organizational perspective—where survey owners build and deploy personalized synthetic respondents—from individual end-user scenarios. This clarification substantially improves the framing of the contribution and makes the intended application scenarios much clearer to the reader.
> > - **Missing and incomplete responses**.
> > In lines 403–405 and Appendix A.2, we now provide detailed statistics on missingness patterns and describe our preprocessing steps with full transparency. This addition improves reproducibility and strengthens the empirical foundation of the paper.
> > - **Comparison with retrieval-based (non-adapter) baselines**.
> > As requested, we added two retrieval-oriented GPT-5 baselines: (i) a profile-only zero-shot setting and (ii) a retrieval-augmented variant that incorporates the top-5 most relevant historical QA exemplars. The results in Tables 1 and 2, together with Relative of Improvments performance in Table 13 and Table 14 show that our adapter-based methods (Aplaud and Aplaud+) siginificantly outperform both retrieval baselines across datasets and both backbones. For example, with the Llama2-7B backbone (Table 1 and  13), Aplaud+ achieves up to +13.9\% higher accuracy (TS), +10.6 \% higher Macro-F1 (TS) compared to GPT5-RAG. Similar patterns hold for Mistral-7B (Table 2 and Table 14), where Aplaud+ surpasses GPT5-RAG by +16.8\% Macro-F1 and +12.3 \% accuracy on TS, and by +2.4\% accuracy, 7.8\% Macro-F1 on average.
> > Lines 477–497 provide a detailed analysis showing why retrieval alone is insufficient: it cannot infer user-specific semantic tendencies, relies heavily on surface-level similarity of prior QA pairs, and fails when relevant exemplars are sparse or non-analogous. In contrast, our personalized adapters learn stable, respondent-specific latent preferences, enabling significantly stronger generalization on unseen questions. These findings underscore the necessity of learned personalization and demonstrate that adapter-based fine-tuning yields a substantial and consistent advantage over best retrieval baselines.
> > -  **Relation to recommender systems**. Appendix F now provides a comprehensive discussion of the conceptual similarities and key differences between our setting and classical recommender-system formulations. This addresses the reviewer’s concerns regarding positioning and helps situate our contribution within existing methodological traditions.
> >
> > In addition to the above revision, we also significantly updated the experimental results part including deeper comparison and analysis on the results, including two new ROI Tables 13 and 14, to highlight our performance superiority across different baselines; extended our evaluation to text general tasks and provided more evidence to validate our assumptions of using shared $(U,V)$ space from SVD. We added several new results from Page 20-24.
> >
> > Overall, these revisions directly and comprehensively address all raised concerns while substantially improving the clarity, empirical rigor, and practical relevance of the work. The reviewer’s suggested additions, particularly the deployment scenario clarification and the retrieval-based baselines, have meaningfully strengthened the paper, and we are grateful for these insights. We hope that the revised manuscript will now be viewed as meeting the acceptance threshold.

---

### Official Review · Reviewer_ukg7 · 2025-10-31

**Soundness:** 3
**Presentation:** 3
**Contribution:** 2
**Rating:** 4
**Confidence:** 3

**Summary:**

The paper tackles personalized survey response prediction with LLMs under three constraints: scarce per-user data, storage/serving scalability, and shared survey structure. It proposes APlaud, which extends LoRA by splitting adaptation into a frozen shared low-rank basis and a compact user-specific correction, further refined by a rank-one residual; the correction can be factorized to cut per-user parameters and overfitting. Experiments (primarily classification) indicate APlaud achieves scalable personalization with improved generalization and inference efficiency over LoRA-based baselines.

**Strengths:**

- The paper conducts extensive experiments on classification tasks across many datasets.
- The focus on time and space efficiency is well-motivated, and the paper includes experiments and analyses demonstrating the corresponding savings.

**Weaknesses:**

- The method is evaluated only on classification tasks. While I understand the work focuses on survey response prediction, the approach should in principle be applicable to generation tasks as well. Notably, OPPU reports results on both generation and classification (e.g., on LAMP). The current evaluation therefore narrows the paper’s contribution.
- The assumption of a stable shared SVD subspace is not strongly validated. It appears that the stage-2 users in training are drawn from the same population as stage-1. Given this coupling in the training data, it is unclear how stage-2 personalization can be cleanly disentangled into the residual component. More evidence is needed to show that the residual truly captures user-specific information rather than leakage from the shared component.
- Users may have different amounts of available data, which could warrant different personalized ranks. The paper lacks fine-grained analysis on how the personalized rank should vary with per-user data volume (and the impact of this choice on performance and overfitting).

**Questions:**

Please see Weakness.

---

> ### Author Response · Authors · 2025-11-23
>
> We thank Reviewer ukg7 for the thoughtful and constructive feedback. We appreciate the clear summary of our work and the recognition of its strengths in scalability and efficiency. The reviewer’s comments on extending evaluation to generation tasks, validating the shared subspace, and adapting rank by data volume are insightful and have helped us refine and clarify the paper’s contributions.
>
> > W1: The method is evaluated only on classification tasks and should be applicable to generation tasks as well.
>
> T1: Generation Task Performance on LaMP News Headline Dataset
> | Method        | R-1    | R-L    |
> |--------------|--------|--------|
> | GPT5-profile | 0.1312 | 0.1177 |
> | OPPU         | 0.1987 | 0.1832 |
> | Cu           | 0.1987 | 0.1825 |
> | Aplaud       | 0.2003 | 0.1838 |
> | Aplaud+      | 0.1987 | 0.1830 |
>
> We thank the reviewer for this helpful comment. To examine whether the proposed personalization mechanism also generalizes to text generation, we further evaluated APlaud on the *LAMP News Headline Generation* dataset. As shown in T1(Table~8 in the manuscript), both **APlaud** and **APlaud+** achieve performance comparable to the strong OPPU baseline while using only about 1% of its per-user parameters. These results indicate that APlaud’s low-rank personalized adaptation extends effectively beyond classification to generative tasks, maintaining efficiency and strong personalization capability.

---

> ### Author Response · Authors · 2025-11-23
>
> > W2: Subspace stability and training data leakage concerns between stage 1 and stage 2
>
> We thank the reviewer for this important concern.
> - **Addressing Leakage Concerns**: To definitively address potential leakage concerns, we designed a strict separation experiment in which Stage 1 pretraining uses an entirely independent $20\%$ subsample of users with zero overlap with any Stage 2 training or test users. This ensures that personalization-stage users contribute nothing to the shared basis learning. As shown in T2 (Table 9 of the revised manuscript), the performance of **APlaud** and **APlaud+** remains virtually unchanged under this setting, demonstrating that Stage 1 captures only general task-level structure, while Stage 2 is solely responsible for learning user-specific adaptations. These findings confirm that our two-stage training cleanly disentangles shared semantic modeling from user-level personalization, and that the personalized factor $C_u$ along with personalized residual $\alpha_u$ and $\beta_u$ indeed capture genuine user-specific variation rather than leaked shared information.
>
> T2: Performance comparison where users in stage-1 do not overlap with ssers in stage-2
> | Method   | G&L | G&L | TS | TS | F&R | F&R | EI | EI | GSS | GSS |
> |---------|:---:|:---:|:---:|:---:|:---:|:---:|:---:|:---:|:---:|:---:|
> |         | ACC | F1 | ACC | F1 | ACC | F1 | ACC | F1 | ACC | F1 |
> | LoRA    | 0.5749 | 0.5344 | 0.6667 | 0.3736 | 0.4929 | 0.2416 | 0.4642 | 0.3687 | 0.3136 | 0.1707 |
> | OPPU    | 0.6667 | 0.6529 | 0.7305 | 0.5995 | 0.6135 | 0.3471 | 0.5042 | 0.3951 | 0.3771 | 0.2909 |
> | Cu      | 0.6441 | 0.6425 | 0.7305 | 0.5987 | 0.6196 | 0.3440 | 0.5208 | 0.3955 | 0.3775 | 0.2943 |
> | Aplaud  | 0.6506 | 0.6506 | 0.7599 | 0.6335 | 0.6265 | 0.3695 | 0.5167 | 0.3927 | **0.4124** | **0.3259** |
> | Aplaud+ | **0.6860** | **0.6717** | **0.7639** | **0.6401** | **0.6291** | **0.3706** | **0.5208** | **0.3981** | 0.3969 | 0.3147 |
>
> - **Empirical Validation of Subspace Stability**:  To directly examine whether our shared-subspace assumption holds, we computed the average (across modules within each layer) *Centered Kernel Alignment* (CKA) similarities of the SVD components $U$ and $V$ between two distinct topics (G\&L vs. TS), as shown in Fig.4 in the revised manuscript. We observe that $V$ maintains high cross-topic stability (CKA $\approx$ 0.96), functioning as a consistent coordinate generator, while $U$ shows moderate but coherent variation (CKA $\approx$ 0.20) that captures topic-specific semantics. This pattern demonstrates that survey data exhibit sufficient structural regularity for a shared low-rank decomposition to remain well-aligned across topics.
>
> - **Domain-specific structural regularity**: Survey data exhibit unique characteristics that naturally support our architectural decomposition of content semantics from user preferences. Survey organizations intentionally design questionnaires with semantically consistent questions across cohorts and waves to serve specific research or marketing objectives. The practice of reusing survey instruments enables comparable measurements over time—a critical requirement since re-contacting respondents is costly and often infeasible. Importantly, these organizations carefully control their sampling frames to avoid mixing unrelated user populations, as respondents outside the intended demographic would invalidate analytic goals. They also recognize time as a key factor, rarely assuming that surveys from previous years represent current conditions in rapidly changing domains.
> Within any given survey, the set of questions presented to users is typically identical or highly overlapping. This structural regularity—where questions are reused but users differ in their responses—directly motivates our architectural separation of *survey content* (modeled via the shared subspace $U, V$) from *user preference* (modeled via personalized components $C_u$, $\alpha_u, \beta_u$). The shared low-rank basis captures content semantics, while personalized residual parameters encode individual response tendencies within this stable semantic framework.
>
> Finally, we would like to address our method does not assume a universal $(U, V)$ pair generalizes across all topics or time periods. In practice, the model learns separate $(U, V)$ bases for each topic, ensuring personalization operates within an appropriately aligned subspace specific to that survey domain. This design acknowledges that different topics (e.g., health vs. political attitudes) may have distinct semantic structures, while still leveraging the within-topic regularity that characterizes well-designed survey instruments.
>
> We appreciate the reviewer’s question, which prompted this deeper analysis and clarification, strengthening the empirical grounding of our approach.

---

> ### Author Response · Authors · 2025-11-23
>
> > W3: Personalized rank should vary with per-user data volume
>
> We thank the reviewer for this thoughtful observation. We address this through three key points: existing ablation analysis, domain-specific justification, and framework extensibility.
>
> - **Existing ablation analysis**: In Appendix~B, our ablation study examines different ranks for both $C_u$ and $(\alpha_u, \beta_u)$. The results show that increasing the rank produces only marginal performance gains while significantly increasing parameter cost, suggesting that the proposed configuration already balances performance and efficiency effectively for the range of data volumes observed in our datasets.
>
> - **Domain-specific justification**: Within the survey-based personalization setting, all users respond to the same or highly overlapping sets of questions, resulting in consistent structural regularity and comparable data volumes across users. Unlike domains where user engagement varies drastically (e.g., e-commerce with sparse vs. power users), survey participants typically provide responses to a standardized question set, which naturally leads to similar amounts of available data per user. Consequently, using a uniform rank across users is both practical and sufficient for the domain considered in this paper.
> Moreover, our empirical results demonstrate that even users with limited responses (e.g., 1–2 answers) benefit from personalization using the fixed-rank configuration, suggesting that the shared low-rank subspace provides effective regularization that prevents overfitting even when per-user data is minimal.
>
> - **Framework extensibility**: That said, our framework is inherently flexible and can naturally accommodate user-specific rank allocation. For instance, one could adaptively set the rank of \(\alpha_u \beta_u^{\top}\) or expand \(P_u Q_u\) based on per-user data volume, uncertainty estimates, or detected distributional shifts. This would be particularly relevant in domains with more heterogeneous user engagement patterns or when handling “atypical” users whose characteristics differ substantially from the main population.  However, such adaptive mechanisms introduce additional considerations:
>    - *Computational Complexity*: Dynamic rank selection requires overhead for per-user rank determination and potentially heterogeneous batch processing.
>   - *Hyperparameter Tuning*: Selecting appropriate rank-allocation strategies (e.g., data-volume thresholds, validation-based selection) adds methodological complexity.
>   - *Overfitting Risk*: Without careful regularization, higher ranks for data-rich users could lead to memorization rather than generalization.
>
> Given that our focus is on standard survey populations with relatively uniform data characteristics—and that our fixed-rank ablation already demonstrates effective performance–efficiency trade-offs—we did not explore adaptive rank mechanisms in the present work. We appreciate the reviewer highlighting this important research direction, which represents a natural extension for applications involving broader user heterogeneity or variable engagement levels.

---

### Official Review · Reviewer_DHHA · 2025-11-01

**Soundness:** 3
**Presentation:** 3
**Contribution:** 3
**Rating:** 4
**Confidence:** 3

**Summary:**

This paper proposes the APlaud method, which aims to address the problem of personalized questionnaire prediction for large models. Through low-rank decomposition combined with user-specific correction, it achieves efficient parameter savings with strong generalization capability. Experiments show that on multiple public social survey and personalized datasets, this method significantly reduces parameters while improving prediction accuracy compared to existing personalization approaches. The code has been open-sourced.

**Strengths:**

The main advantages of this paper are: First, it significantly reduces the number of personalized parameters per user compared to existing methods such as OPPU, making large-scale personalized modeling feasible;

Second, by combining a shared low-rank subspace with user-specific residuals, it not only saves memory but also maintains or even improves the generalization accuracy of personalized models, outperforming existing personalized LoRA methods on multiple public datasets.

**Weaknesses:**

The main limitations of this paper are:
1. The method's ability to express personalized residuals is highly dependent on the structural choice of low-rank decomposition. If user feature distributions vary significantly, the low-rank subspace may struggle to comprehensively cover all user needs, affecting performance in extreme personalization scenarios.

2. The experiments mainly focus on questionnaire and text-based personalization tasks. The generalization capability for more complex, multimodal, or multi-turn deep personalization scenarios requires further validation.

**Questions:**

In extreme personalization scenarios or with long-tail users (whose characteristics differ significantly from mainstream users), does the low-rank representation capability of the APlaud method experience substantial degradation? Have you considered implementing automatic detection of "atypical" users and dynamically adjusting the model architecture accordingly?

While the current method primarily focuses on achieving high accuracy with minimal personalized parameters, how do you ensure user data security and privacy protection?

---

> ### Author Response · Authors · 2025-11-23
>
> We sincerely thank Reviewer DHHA for the thoughtful and constructive feedback. We appreciate the clear
> summary of our contributions and the recognition of APlaud’s strengths in reducing per-user parameters and
> improving generalization across personalized LLM tasks. Your comments regarding the limits of low-rank
> representation, the scope of our evaluation, and the importance of privacy have been very helpful. We have
> carefully addressed each of these points in our responses below and believe the resulting clarifications further
> strengthen the paper’s motivation, scope, and practical relevance.
>
> > W1 + Q1: Low-rank subspace may degrade for atypical or long-tail users.
>
> We thank the reviewer for this thoughtful question. We address the concern through three key points:
> - **Domain-specific design considerations**: Our method specifically targets \textit{survey question–answer (QA) prediction}, where question formats, semantic structures, and answer spaces exhibit high consistency across users. Unlike general personalization tasks with arbitrary user diversity, survey data inherently possesses structural regularity—the primary modeling challenge is capturing nuanced user-specific preferences within this stable framework rather than accommodating unbounded heterogeneity.
> - **Architectural separation of shared and personalized factors**: APlaud explicitly addresses this through its decomposition: the shared low-rank subspaces \(U\) and \(V\) encode common semantic factors across survey questions, while user-specific variations are isolated in the personalized correction $C_u$ and residual term $\alpha_u \beta_u^{\top}$. This design allows efficient modeling of individual response tendencies without requiring the low-rank subspace to simultaneously capture both universal patterns and extreme individual variations. While we did not implement dynamic rank expansion in this paper, it is theoretically straightforward to extend APlaud by allocating more personalized residual factors (e.g., higher-rank $\alpha_u \beta_u^{\top}$ or expanded $P_u Q_u$) for users exhibiting atypical or extreme preference behaviors.
> - **Scope and practical context**: Survey organizations typically follow structured designs—administering semantically related questions to preselected, well-characterized participant cohorts. Since re-contacting respondents is costly or often infeasible, our work focuses on predicting follow-up responses *within* the same or closely related survey domains. While extending APlaud to handle extreme cross-population heterogeneity or broader digital-twin scenarios is an interesting future direction, such applications involve fundamentally different modeling assumptions and data characteristics that fall outside the scope of this paper.
>
> > W2: Experiments mainly focus on questionnaire and text-based personalization tasks.
>
> We appreciate the reviewer's interest in broader applications. However, our work deliberately targets \textit{survey-style QA personalization}—that is, predicting individual users’ responses to structured survey questions. This domain is already substantial in scale, economically significant, and widely adopted across both academia and industry, and the problem itself is highly non-trivial. As documented in our Introduction, survey instruments (e.g., Pew Research, General Social Survey, Gallup, ANES) and synthetic survey participants have become foundational tools in marketing research, social science, product design, and policy analysis. Improving predictive personalization in this context can substantially reduce survey costs, improve respondent modeling, and enhance the quality of population-level inferences. Despite its importance, personalization for survey QA remains largely understudied in the machine learning literature. Our work directly addresses this concrete, high-impact challenge through methods specifically designed for survey data’s unique characteristics—structured question formats, semantic consistency, and the need for interpretable, user-specific modeling.
>
> Extending APlaud to multimodal or multi-turn personalization scenarios is an interesting direction for future work, but such extensions would require different architectural and evaluation frameworks. Establishing a strong foundation in survey personalization is, in our view, a necessary step toward building more general, scalable personalization systems.

---

> ### Author Response · Authors · 2025-11-23
>
> > Q2: How to ensure user data security and privacy protection?
>
> We thank the reviewer for raising this important consideration. Our method incorporates several design choices that inherently support privacy protection:
>
> - **Compact latent representations**:  APlaud represents each user through compact latent parameters. Specifically, the personalized correction matrix $C_u$ and residual factors $(\alpha_u, \beta_u)$. Once these parameters are learned, predictions require only these low-dimensional representations rather than repeatedly accessing or transmitting raw user response histories. This design minimizes the exposure of sensitive individual-level data during inference.
> Furthermore, unlike retrieval-augmented or memory-based personalization methods that repeatedly query, move, or expose user records across system components, our parametric approach encapsulates user-specific information within model parameters. This substantially reduces the risk of data leakage or unauthorized access during deployment.
> - **Domain context and existing protections**: Our work focuses on survey data, which is typically governed by institutional privacy protocols. Many widely used survey datasets (e.g., Pew Research, GSS, Gallup) are anonymized at collection and managed under strict data governance policies. APlaud operates within this existing privacy infrastructure and does not require additional sensitive data collection beyond what survey providers already manage.
> - **Enterprise-level privacy controls**: In practical deployment, additional safeguards are typically enforced at the enterprise or service-provider level. For instance, user-specific parameters or embeddings must remain isolated to prevent cross-contamination between users. This isolation is commonly implemented in modern LLM service infrastructures such as vLLM, which provides automatic key–value (KV) cache cleanup and sandboxed execution for independent inference sessions. Such practices complement our compact parametric design to ensure end-to-end protection across both training and inference stages.
>
> Finally, while our current design offers these inherent privacy advantages, incorporating formal privacy guarantees—such as differential privacy during fine-tuning—represents a valuable direction for future work, particularly for deployment in sensitive or regulated domains. We believe our compact parametric representation provides a strong foundation for such privacy-preserving extensions.

---

### Author Response · Authors · 2025-11-24
**Thank you for the thoughtful and constructive reviews!**

We sincerely thank all reviewers for their thoughtful, constructive, and detailed feedback. In response to your suggestions and concerns, we have made the following key additions and clarifications to improve the paper:

- In line 392-line 395 and Appendix A.2 we added more detailed descriptions on missing and incomplete data statistics and how we handle them during pre-processing, as suggested by reviewer **oHND**.

- In line 404 - line 407, together with revised Table 1 and
Table 2, we added GPT-5 as zero shot baseline and also a retrieval-based baseline follwing suggestion by reviwer **oHND**. These results show that memory-based retrieval alone is inadequate for robust personalization, whereas APlaud’s structured low-rank personalized adaptation delivers significantly stronger and more consistent performance gains across datasets.

- We correct a typo in line 920, thanks to comments by reviwer **broL**

- We added a detailed description on how we obatined user profile in Appendix D.3 and conducted additional experiments to compare show how Aplaud performs without profile information in the prompt in Appendix C.2 and Table 7, as suggested by reviwer **broL**. The results demonstrated that our lightweight parametric modules effectively capture stable user preference patterns directly from response histories without requiring explicit profile features.

- In response to reviwer **ukg7** and **oHND**, we clarified our deseign scope and  extended experiments to evaluate Aplaud on text generation task, as added in Table 8 and Appendix C.3, These results indicate that APlaud’s low-rank personalized adaptation extends effectively beyond classification to generative tasks, maintaining efficiency and strong personalization capability.

- We validated our assumption that shared $(U, V)$ would capture general task space knowledge and measure the subspace similarity across different topics leveraging CKA similarity in Appendix C.7 and Fig.4 as suggested by reviewer **broL** and **DHHA**. In response to reviewer **ukg7**, we additionally confirm that our two-stage training cleanly disentangles shared semantic modeling from user-level personalization by designing a strict separation experiment in which Stage 1 pretraining uses an entirely independent subsample of users with zero overlap with any Stage 2 training or test users in Table 9 and Appendix C.4.

- In response to reviewer **oHND**, we added a relation with recommendation system in Appendix.F and clarified our novelty in this paper.

We again thank all reviewers for their insightful feedback. Please let us know if any of your concerns remain unaddressed. We will update the revision accordingly.

---

> ### Author Response · Authors · 2025-12-04
>
> =====================================UPDATES=========================================
>
> Thanks to the reviewers for their thoughtful comments and engagement. We have uploaded a new revision that includes additional discussions and clarifications. Building upon the previous version, we have made the following updates:
>
> - In lines 33–37, 70–78, and 131–137, we clarified the real-world deployment setting and real-world motivation of our work as suggested by reviewer **OHND**.
>
> - In lines 402–406 and Appendix A.2, we included a thorough description of missing and incomplete data statistics and how we handle them during pre-processing, as suggested by reviewer **oHND**.
>
> - We added a detailed description on how we obtained the user profile in Appendix D.3 and conducted additional experiments to show how Aplaud performs without profile information in the prompt in Appendix C.2 and Table 7, as suggested by reviewer **broL**. The results show that Aplaud still outperforms OPPU by a large margin, demonstrating that our lightweight parametric modules effectively capture stable user preference patterns directly from response histories without requiring explicit profile features.
>
> - In response to reviewers **ukg7** and **oHND**, we clarified our design scope and extended experiments to evaluate Aplaud on text generation task, as added in Table 8 and Appendix C.3, These results indicate that APlaud’s low-rank personalized adaptation extends effectively beyond classification to generative tasks, maintaining efficiency and strong personalization capability.
>
> - We validated our assumption that shared would capture general task space knowledge and measure the subspace similarity across different topics, leveraging CKA similarity in Appendix C.7 and Fig.4 as suggested by reviewer **broL** and **DHHA**. In response to reviewer **ukg7**, we additionally confirm that our two-stage training cleanly disentangles shared semantic modeling from user-level personalization by designing a strict separation experiment in which Stage 1 pretraining uses an entirely independent subsample of users with zero overlap with any Stage 2 training or test users in Table 9 and Appendix C.4.
>
>
> - Appendix F now provides a comprehensive discussion of the conceptual similarities and key differences between our setting and classical recommender-system formulations. This
> addresses the reviewer **oHND**'s concerns regarding positioning and helps situate our contribution within existing
> methodological traditions
>
>
> - We added two retrieval-oriented GPT-5 baselines: (i) a profile-only zero-shot setting and (ii) a retrieval-augmented variant as suggested by reviewer **oHND**. We included results in revised Table 1 and Table 2 and added Table 13 and Table 14 to highlight the superior personalization performance of our approaches (in terms of ROI) over various baselines across datasets. Typically, our approach achieves up to 20.6\% higher accuracy, 40.3\% higher Macro-F1 compared to GPT5-zero-shot; 2.4\% Accuracy and 7.8\% Macro-F1 over the retrieval-based baseline,
> and by +2.5\% accuracy, 10.2\% Macro-F1 over the strong personalized baseline OPPU on average. We also provide a deeper analysis explaining why retrieval alone is insufficient in lines 477-497.
>
> - We correct a typo in line 920, thanks to comments by Reviewer **broL**.
>
> We acknowledge that reviewers can no longer respond, and we hope these updates adequately address all their concerns.

---

### Meta-Review · Area_Chair_Wphe · 2026-01-01

**Summary:**

The paper proposes APlaud, a parameter-efficient personalization method decomposing adaptation into shared and user-specific components, but is recommended for rejection due to consensus concerns regarding the limited evaluation scope (primarily structured surveys), the method's questionable advantage over strong retrieval-based baselines, and unresolved limitations in handling highly heterogeneous or long-tail user distributions.

**Reviewer Concerns:**

While the rebuttal successfully addressed requests for generative task evaluation (LaMP) and retrieval baselines (GPT-5), critical concerns regarding the method's practical utility over simpler retrieval approaches and its ability to robustly handle atypical user profiles remain outstanding.

**Reviewer Scores:**

Reviewers DHHA and ukg7 would likely have slightly improved their scores had they fully engaged with the latest generative task results which addressed their generalization concerns, while Reviewer oHND, despite raising their score to weak rejection (4) during the discussion, likely would have remained at the acceptance threshold due to persisting doubts about the real-world necessity of the method compared to retrieval-augmented generation.

---

### Decision · Program_Chairs · 2026-01-26

Reject